# Cryo-electron microscopy structure of an archaeal ribonuclease P holoenzyme

Futang Wan[1,2,3,8], Qianmin Wang[4,5,8], Jing Tan[1,2,8], Ming Tan [1,2], Juan Chen[4,5], Shaohua Shi[4,5], Pengfei Lan [4,5], Jian Wu[4,5,6] & Ming Lei [4,5,7]

Ribonuclease P (RNase P) is an essential ribozyme responsible for tRNA 5′ maturation. Here we report the cryo-EM structures of *Methanocaldococcus jannaschii* (*Mja*) RNase P holoenzyme alone and in complex with a tRNA substrate at resolutions of 4.6 Å and 4.3 Å, respectively. The structures reveal that the subunits of *Mja*RNase P are strung together to organize the holoenzyme in a dimeric conformation required for efficient catalysis. The structures also show that archaeal RNase P is a functional chimera of bacterial and eukaryal RNase Ps that possesses bacterial-like two RNA-based anchors and a eukaryal-like protein-aided stabilization mechanism. The 3′-RCCA sequence of tRNA, which is a key recognition element for bacterial RNase P, is dispensable for tRNA recognition by *Mja*RNase P. The overall organization of *Mja*RNase P, particularly within the active site, is similar to those of bacterial and eukaryal RNase Ps, suggesting a universal catalytic mechanism for all RNase Ps.

[1] State Key Laboratory of Molecular Biology, CAS Center for Excellence in Molecular Cell Science, Shanghai Institute of Biochemistry and Cell Biology, Chinese Academy of Sciences, Shanghai 200031, China. [2] University of Chinese Academy of Sciences, Chinese Academy of Sciences, Shanghai 200031, China. [3] School of Life Science and Technology, ShanghaiTech University, Shanghai 201210, China. [4] Ninth People's Hospital, Shanghai Jiao Tong University School of Medicine, Shanghai 200125, China. [5] Shanghai Institute of Precision Medicine, Shanghai 200125, China. [6] Shanghai Key Laboratory of Translational Medicine on Ear and Nose diseases, Shanghai 200125, China. [7] Key laboratory of Cell Differentiation and Apoptosis of Chinese Ministry of Education, Shanghai Jiao Tong University School of Medicine, Shanghai 200025, China. [8]These authors contributed equally: Futang Wan, Qianmin Wang, Jing Tan. Correspondence and requests for materials should be addressed to P.L. (email: pengfeilan@shsmu.edu.cn) or to J.W. (email: wujian@shsmu.edu.cn) or to M.L. (email: leim@shsmu.edu.cn)

Ribonuclease P (RNase P) and ribosome are the only two naturally occurring ribozymes that are present in all three kingdoms of life. RNase P is an ancient ribonucleoprotein (RNP) complex that catalyzes the maturation of the 5′ end of precursor tRNAs (pre-tRNAs)[1–3]. Bacterial RNase P is composed of a single small RNase P protein (RPP) in addition to the RNase P RNA (RPR)[4]. Based on the secondary structures, bacterial RPRs have been further classified into two subtypes, the most common type A for ancestral and type B for Bacillus[5]. Structural information of both bacterial A-type and B-type RPRs reveals that RPR consists of two independently folded domains, the catalytic domain (C-domain) and the specificity domain (S-domain), which play key roles in substrate cleavage and substrate binding, respectively[6–11]. It has been found that bacterial RPR even the C domain alone possesses catalytic activity under high ionic strength condition or in the presence of spermidine in vitro[12,13]. But the sole protein subunit RPP is essential for enhancing the efficiency and fidelity of substrate recognition and cleavage under physiological conditions[14–19].

In contrast to bacteria, archaeal and eukaryal nuclear RNase Ps have evolved considerably more complex protein subunits, four to five in archaeal and nine to ten in eukaryal enzymes[2]. Archaeal RNase P is an evolutionary intermediate with chimeric features of both bacterial and eukaryal nuclear enzymes, and thus serves as an excellent system to provide insights into the structural and functional alterations that accompanied the gradual transformation of an ancient catalytic RNA to a protein-rich RNP[3,20]. Archaeal RNase Ps can be further classified into three major types (A, M, and P) on the basis of the secondary structural characteristics of their RPRs[21–23]. A-type RPR largely resembles bacterial RPRs and displays trace amount of catalytic activity in vitro, whereas M-type RPR diverges more from bacterial RPRs with less complex structure and has not been shown the ability as an RNA-only ribozyme[24,25]. Type P is a more extreme case in which the RPR is only about two thirds of its type A and M relatives, but surprisingly the RPR itself still retains the tRNA processing activity in vitro[21].

Previous biochemical studies of in vitro reconstituted archaeal RNase Ps provide us the first glimpse of the roles of RPP subunits[24,26,27]. Single turn-over kinetic studies revealed that protein subcomplex (Pop5-Rpp30)$_2$ is responsible for enhancing the cleavage rate of the tRNA precursors, while Rpp21-Rpp29 contributes to the increased substrate affinity[28]. Ribosomal protein L7Ae is the fifth subunit of archaeal RNase P and was shown to increase the thermostability of the Pyrococcus horikoshii RNase P holoenzyme[29,30]. Although crystal structures of individual protein subunits and protein subcomplexes have been determined[29,31–38], their structural and enzymatic roles in archaeal RNase P holoenzyme still remain unclear.

Recently, mechanistic understanding of eukaryal RNase P has been taken a big step forward through the cryo-electron microscopy (EM) structures of the yeast and human RNase Ps, which reveal the spatial organization of eukaryal nuclear RNase Ps as well as the detailed mechanisms of substrate recognition and processing[39,40]. To gain insights into the structure and function of archaeal RNase P holoenzyme and its evolutionary relationships with bacterial and eukaryal enzymes, we reconstitute the RNase P holoenzyme from Methanocaldococcus jannaschii (M-type) and determine its cryo-EM structure alone and in complex with a tRNA substrate. The structures fill a void for the structural insights into the RNase P evolution and provide mechanistic understanding of the catalysis of archaeal RNase P.

## Results

### In vitro reconstitution of the *Mja*RNase P holoenzyme. The *Mja*RNase P holoenzyme contains a 252-nucleotide RPR and

five protein subunits Pop5, Rpp30, Rpp29, Rpp21, and L7Ae[28,30,41]. To reconstitute the *Mja*RNase P complex, we first over-expressed L7Ae as an individual subunit, Rpp29-Rpp21 and (Pop5-Rpp30)$_2$ as heterodimeric and heterotetrameric subcomplexes in *Escherichia coli*, respectively (Supplementary Fig. 1a). The *Mja*RPR was transcribed and purified from an in vitro transcription system. However, RPR alone behaved poorly and tended to form soluble aggregates as revealed by size exclusion chromatography (SEC) analysis (Supplementary Fig. 1b). Notably, when purified L7Ae was added into the transcription reaction, RPR behaved properly as a mono-dispersed molecule, suggesting that L7Ae presumably functions as a chaperone for the correct folding and stability of RPR (Supplementary Fig. 1b). The in vitro transcribed RPR in the presence of L7Ae was sequentially mixed with purified Rpp29-Rpp21 and (Pop5-Pop30)$_2$ subcomplexes and analyzed by SEC (Fig. 1a). Strikingly, addition of (Pop5-Rpp30)$_2$ shifted the elution peak of the holoenzyme to a position with an apparent molecular weight of ~440 kDa that is about twice of the calculated molecular weight of *Mja*RNase P (~210 kDa) (Fig. 1a), suggesting that the reconstituted *Mja*RNase P likely adopts a dimeric configuration that is mediated by the (Pop5-Rpp30)$_2$ heterotetramer. Next, we employed negative staining EM to further examine the oligomeric state of the *Mja*RNase P complex, and found that the homogenous and monodispersed particles exhibited an elongated overall conformation (Fig. 1b). Consistent with the SEC analysis, two-dimensional class average of the particles confirmed that the reconstituted *Mja*RNase P holoenzyme is indeed a dimeric complex with a clear two-fold symmetry (Fig. 1b).

Careful analysis of the *M. jannaschii* genome identified 35 tRNA genes (Supplementary Table 1). Incubation of *M. jannaschii* pre-tRNA$^{Arg}$ with the reconstituted *Mja*RNase P holoenzyme leds to the cleavage of the 5′ leader from pre-tRNA$^{Arg}$ molecule in a magnesium ion (Mg$^{2+}$) dependent manner (Fig. 1c and Supplementary Fig. 2a, b). This result demonstrated that the in vitro reconstituted *Mja*RNase P complex is a fully functional enzyme. Notably, *Mja*RNase P can also efficiently process both human and *E. coli* pre-tRNAs (Supplementary Fig. 2a, b), consistent with the notion that RNase P recognizes the conserved structural feature, but not specific sequences of tRNA molecules[39,40].

**Overall architecture of *Mja*RNase P**. To reveal the structure of *Mja*RNase P, the reconstituted *Mja*RNase P holoenzyme was subjected to cryo-EM analysis using a Falcon III direct camera, resulting in a well-defined electron density map of *Mja*RNase P at a resolution of 4.6 Å (Fig. 2a, Supplementary Fig. 3 and Supplementary Table 2). To further understand how the tRNA substrate is recognized and processed by *Mja*RNase P, we mixed *Mja*RNase P with *E. coli* pre-tRNA$^{Tyr}$ at a ratio of 1:10 and subjected the mixture to cryo-EM single particle analysis. Notably we obtained the three-dimensional reconstruction of *Mja*RNase P in complex with the mature form of the tRNA$^{Tyr}$ substrate at a resolution of 4.3 Å (Fig. 2a, Supplementary Fig. 4 and Supplementary Table 2). We speculated that the 5′ leader of pre-tRNA$^{Tyr}$ was cleaved during EM sample preparation.

The EM reconstruction revealed that the *Mja*RNase P holoenzyme indeed adopts a dimeric conformation (Fig. 2a). The resolutions of the cryo-EM reconstruction were substantially improved by applying the two-fold symmetry, suggestive of a very rigid dimeric interface in the *Mja*RNase P complex (Supplementary Figs 3, 4). Secondary structural elements were clearly resolved in the EM density maps, allowing all folded domains

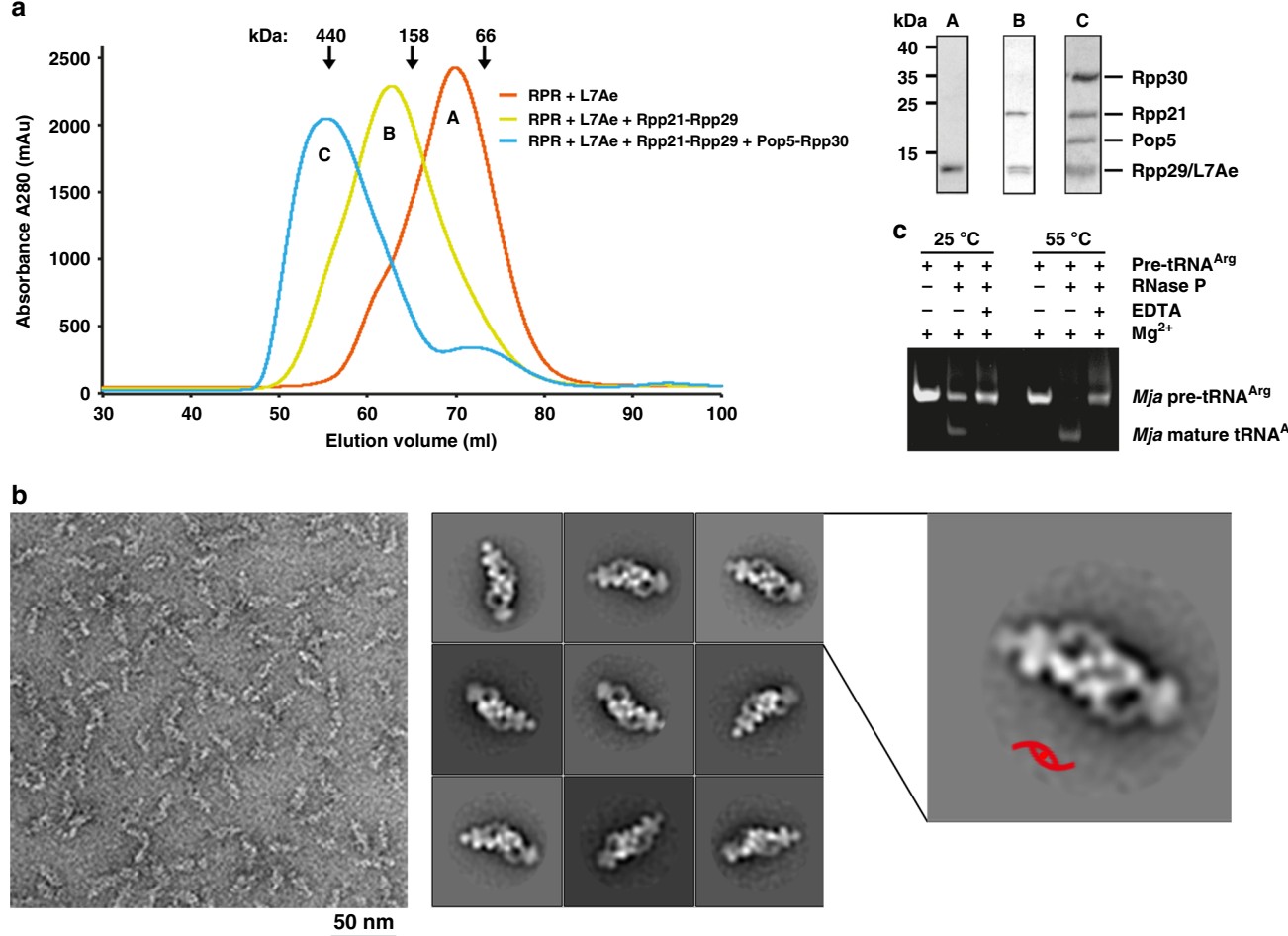

**Fig. 1** In vitro reconstitution of the *Mja*RNase P holoenzyme. **a** Size exclusion chromatographic profiles (left) and Coomassie-stained SDS-PAGE of the corresponding peaks in the profiles (right). **b** Negative staining EM analysis of *Mja*RNase P. Left: A representative negative-stained EM micrograph of *Mja*RNase P. Middle: Selected 2D class averages of *Mja*RNase P. Right: Close-up view of the 2D class averages with the C2 symmetry denoted a red symbol. **c** In vitro pre-tRNA$^{Arg}$ processing assay of the *Mja*RNase P holoenzyme

predicted within *Mja*RNase P to be assigned (Supplementary Fig. 5). By using homologous structure modeling based on the crystal structures of bacterial RPR from *Thermotoga maritima*, archaeal components from *P. horikoshii*, and the cryo-EM structures of both yeast and human RNase P complexes, we generated an atomic model of the dimeric *Mja*RNase P holoenzyme, containing ten protein subunits and two catalytic RPRs (Fig. 2b).

The protein components of the *Mja*RNase P complex are arranged into an elongated configuration, which can be divided into three submodules, one (Pop5-Rpp30)$_2$ heterotetramer at the center and two Rpp29-Rpp21-L7Ae heterotrimer at the ends (Fig. 2b). These protein submodules are intimately stringed together and serve as a long, extended holder to accommodate the two RPRs through a highly basic surface (~6500 Å$^2$) (Fig. 2c). The (Pop5-Rpp30)$_2$ heterotetramer mediates symmetric interactions with two RPR molecules, which are organized into an extended, twisted Z-shaped configuration (Fig. 2b). Both termini of this extended RNA architecture associate with the Rpp29-Rpp21-L7Ae heterotrimer (Fig. 2b). The structure of the *Mja*RNase P-tRNA complex shows that *Mja*RNase P adopts the same dimeric conformation as the apo structure with two tRNA substrates bound into each of the two active sites (Fig. 2b). Superposition analysis revealed that binding of the tRNA substrate only induces a ~8° change in the angle between the

two *Mja*RNase P monomers, suggestive of a rigid dimeric *Mja*RNase P architecture (Fig. 2d).

**M. jannaschii RPR**. Similar to bacterial and eukaryal RPRs, *Mja*RPR can also be divided into two independently folded domains, the C and the S domains, with limited connections in between (Figs 3a, b)[22,41]. The C domain is composed of stems P1, P2, P3, P4, P5 and P15, and the S domain contains stems P7, P10, P12, P12.1, and P12.2 (Fig. 3a, b). The C and S domains are connected by the P5-P7 stems, and by the interaction between stems P1 and P9 (Fig. 3b). Universally conserved regions CR-I, CR-IV, and CR-V encompassed by stems P1, P2, P3, and P15 form the catalytic pseudo-knot motif (P4), occupying the center of the C domain (Fig. 3c). The other two conserved regions CR-II and CR-III between stems P10 and P12 fold into two interleaved T-loops for tRNA substrate binding (Fig. 3c)[10,39,40,42]. Consistent with previous predictions, both pseudo-knot and T-loop motifs are identical to those observed in bacterial and eukaryal RPRs (Supplementary Fig. 6), confirming that they are conserved structural features in RNase Ps among all three domains of life[10,39,40]. *Mja*RPR also contains an extension beyond stem P12, which folds back to form a characteristic kink-turn (K-turn) structure (Fig. 3b, c)[30]. Phylogenetic and secondary structural analyses revealed that an extension with a K-turn beyond stem

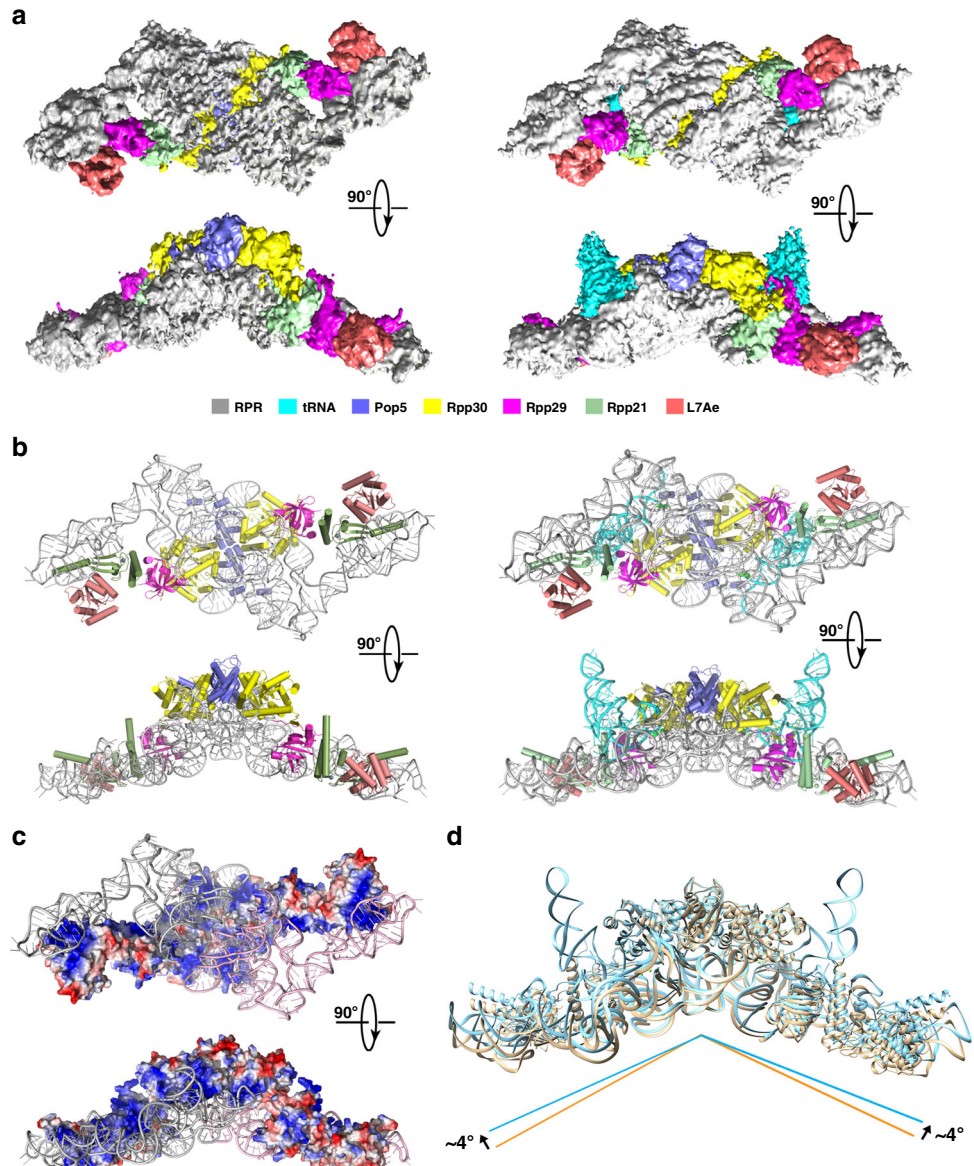

**Fig. 2** Overall structures of the *Mja*RNase P complex with or without tRNA. **a** The cryo-EM density maps of *Mja*RNase P (left) and the *Mja*RNase P-tRNA^Tyr complex (right) are shown in two orthogonal views. Protein and RNA components are color-coded and the scheme is shown below the figure. **b** Two orthogonal views of the atomic model of *Mja*RNase P (left) and the *Mja*RNase P-tRNA complex (right) are shown in cartoon representation. Protein and RNA components are color-coded as in (**a**). **c** Two orthogonal views of the surface electrostatic potential of the protein assembly in *Mja*RNase P reveals a continuous highly basic surface that binds two RPRs (negative: red; positive: blue). The two RPRs are colored in pink and gray, respectively. **d** Superposition of the structures of *Mja*RNase P with or without tRNA shows that tRNA binding only induces a ~8° change in the angle between the two *Mja*RNase P monomers. *Mja*RNase P and the *Mja*RNase P-tRNA complex are colored in wheat and palecyan, respectively

P12 is a conserved feature in most archaeal RNase P RNAs (Supplementary Fig. 7)[30,37,38,43,44].

***Mja*RNase P RPPs.** Crystal structures of all protein components of *P. horikoshii* RNase P have been determined[29,31–34,36]. Given the high sequence conservation between *P. horikoshii* and *M. jannaschii* RPPs (Supplementary Fig. 8), it is not surprising that the structures of individual protein components, the (Pop5-Rpp30)$_2$ heterotetramer and the Rpp29-Rpp21 heterodimer revealed in the *Mja*RNase P holoenzyme structure highly resemble previously determined *P. horikoshii* crystal structures (Supplementary Fig. 9). However, what is surprising is that in the holoenzyme one (Pop5-Rpp30)$_2$ heterotetramer and two

Rpp29-Rpp21-L7Ae heterotrimer are interlinked together to form a long extended decamer with a highly positively charged surface that holds two copies of *Mja*RPR (Figs 2c, 4a, b). A salient feature of this decamer architecture is the symmetric connection between the heterotetramer and two heterotrimers (Fig. 4a). The C-terminal long tail of Rpp30 sticks out of the main body of Rpp30 and folds into a β strand to become the edge of the β barrel of Rpp29, defining the key linkage in the protein decamer (Fig. 4c and Supplementary Fig. 10a). Moreover, the C-terminal short hydrophobic tail of Rpp29 extends out and fits into a hydrophobic groove of Rpp30, further strengthening the connection between the (Pop5-Rpp30)$_2$ heterotetramer and the Rpp29-Rpp21-L7Ae heterotrimer (Fig. 4d and Supplementary Fig. 10b). Notably, except for a few A-type archaeal RNase Ps, the

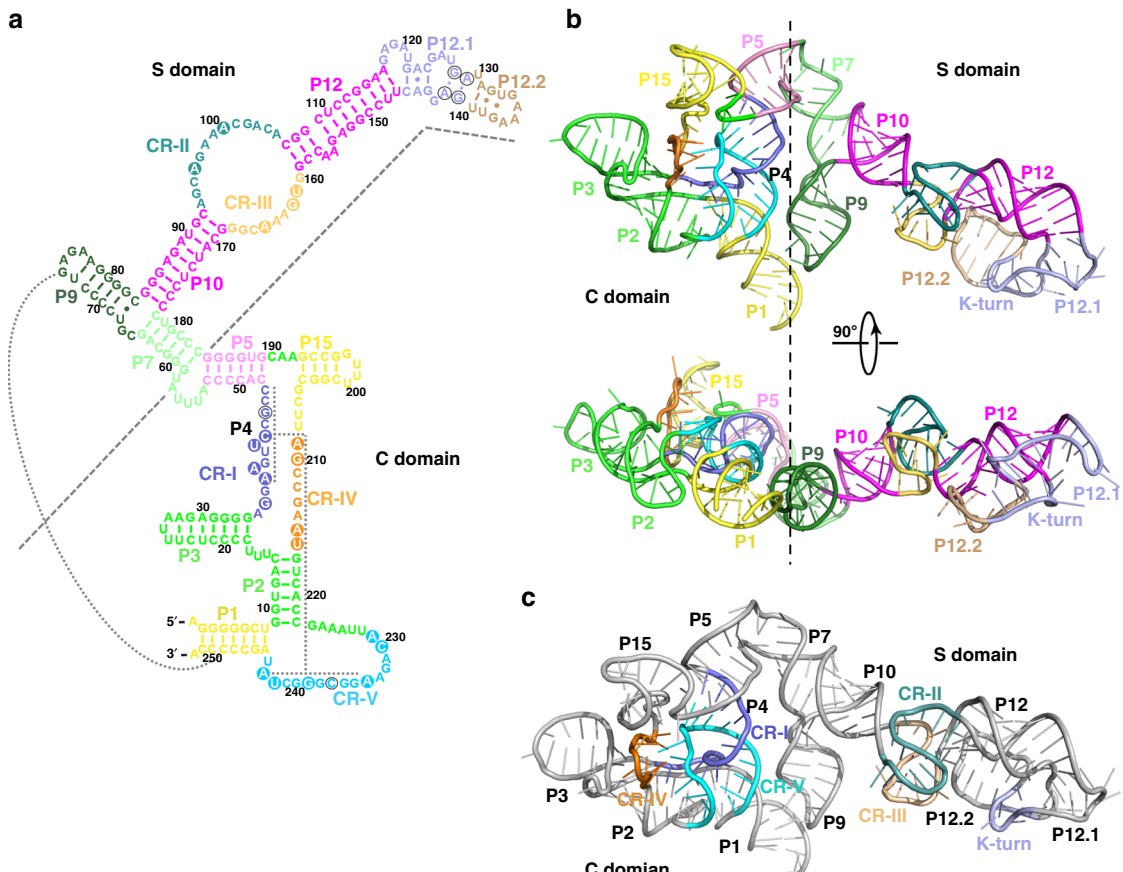

**Fig. 3** Structure of the *M. jannaschii* RPR. **a** Secondary structure of RPR. Structural elements are colored to match their labels. The conserved regions of RPR (CR-I to CR-V) are colored in slate, deepteal, yelloworange, orange and cyan, respectively. The conserved P4 stem is denoted with dotted line. Dotted line between stem P1 and P9 indicates long-range RNA-RNA interaction. **b** Two orthogonal views of the overall structure of RPR. RPR is colored as in (**a**). **c** Overall structure of the RPR. The pseudoknot and the T-loop regions locate in the C and S domains of RPR, respectively. CR-I to CR-V are colored as in (**a**)

C-terminal tail of Rpp30 is highly conserved in most archaeal Rpp30 proteins (Supplementary Fig. 11), suggesting that the intermolecular connection between Rpp30 and Rpp29, and therefore, the overall architecture of the protein assembly is very likely conserved in most archaeal RNase P holoenzymes.

Another previously unobserved protein interaction is between Rpp21 and L7Ae. The L-shaped Rpp21 resides in the middle between Rpp29 and L7Ae, and the three proteins sequentially packs against one another to form the heterotrimer (Fig. 4c and Supplementary Fig. 12a). On one side, Rpp21 mediates extensive interactions with Rpp29 in a similar manner as their *P. horikoshii* homologs (Fig. 4c and Supplementary Fig. 9b)[36]. On the other side, the flat surface of Rpp21 holds L7Ae through both hydrophobic and electrostatic interactions so that L7Ae is in a suitable position to recognize the K-turn of *Mja*RPR (Fig. 4c and Supplementary Fig. 12b, c). The two interfaces bury exposed surface areas of 960 Å[2] and 690 Å[2] respectively, suggesting that the Rpp29-Rpp21 interaction is stronger than that between Rpp21 and L7Ae. Given both Rpp21 and L7Ae interact with *Mja*RPR, it is likely that the protein-RNA interactions assist the formation of the Rpp29-Rpp21-L7Ae heterotrimer in the holoenzyme (Supplementary Fig. 12a). This is consistent with the fact that previous biochemical analysis failed to detect the interaction between Rpp21 and L7Ae[45,46].

In accordance with previous predictions, the *Mja*RNase P holoenzyme structure confirmed that *M. jannaschii* protein components are indeed structural homologs of their eukaryal counterparts (Supplementary Fig. 9a–c). It is noteworthy that archaeal Pop5 had evolved into two different eukaryal proteins

(Pop5 and Rpp14 in human RNase P) (Supplementary Fig. 9a)[47]. Consequently, the *M. jannaschii* (Pop5-Rpp30)₂ heterotetramer became a Pop5-Rpp14-(Rpp30)₂ heterotetramer with a 1:1:2 stoichiometry (Supplementary Fig. 9a)[39]. In addition, not only individual protein components are conserved, the inter-subcomplex (Pop5-Rpp30)₂-(Rpp29-Rpp21-L7Ae) connection mediated by Rpp30 and Rpp29 is also conserved in human RNase P holoenzyme structure (Fig. 4e)[39]. In contrast, none of the *Mja*RNase P protein components is structurally similar to the sole protein subunit RPP of bacterial RNase P (Supplementary Fig. 9a–d), suggesting that *Mja*RPPs are evolutionarily related to eukaryal RPPs, but not to bacterial RPP[47].

**RPP-RPR interactions**. The (Pop5-Rpp30)₂ heterotetramer locates in the center of the dimeric *Mja*RNase P holoenzyme (Fig. 5a). Pop5 adopts a typical RNA-recognition-motif fold, sitting on the junction between stems P2 and P3 of RPR (Fig. 5a). A highly basic cleft formed by the N-terminal tail and C-terminal helices α3 and α4 of Pop5 tightly holds the zigzagged CR-IV of RPR (Fig. 5a). While Pop5 only contacts one RPR in the complex, Rpp30 is involved in electrostatic interactions with both RPR molecules (Fig. 5b). Together, the symmetric contacts between Rpp30 and RPR stems P2-P3 and between Pop5 and CR-IV of RPR mold *Mja*RNase P into a dimeric holoenzyme complex (Fig. 5a, b).

At the distal ends of the *Mja*RNase P complex, the heterotrimer Rpp29-Rpp21-L7Ae associates with the terminal regions of stems P1 and P9 of RPR, and the highly basic Rpp21

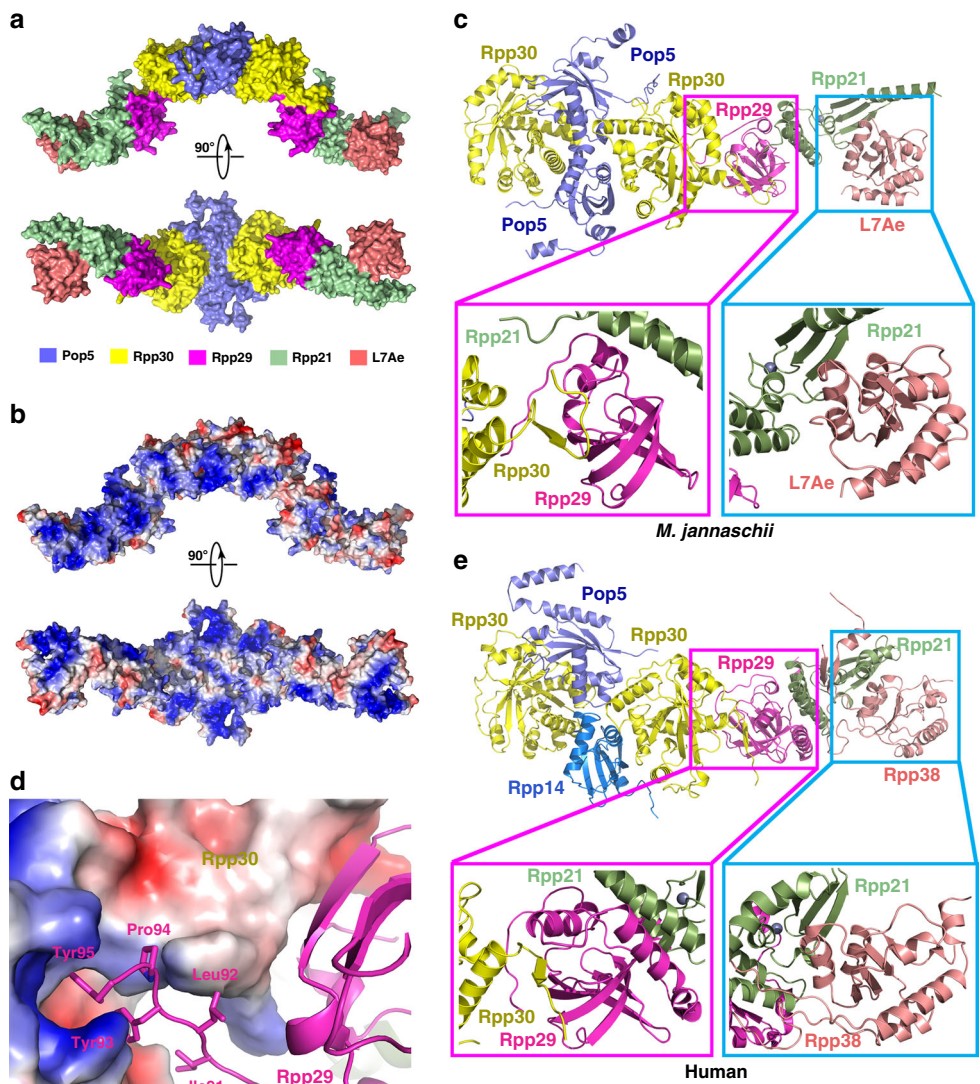

**Fig. 4** Protein components of *Mja*RNase P. **a** Two orthogonal views of the *Mja*RNase P protein assembly is shown in surface representation. Protein components are color-coded and the scheme is shown below the figure. **b** Two orthogonal views of the electrostatic surface potential of the protein assembly (negative: red; positive: blue). **c** Cartoon representation of the *Mja* (Pop5-Rpp30)₂-Rpp29-Rpp21-L7Ae heptamer. Protein components are colored as in (**a**). Bottom left: close-up view of the interface between Rpp30 and Rpp29. Bottom right: close-up view of the interface between Rpp21 and L7Ae. **d** The C-terminal tail of Rpp29 binds into a hydrophobic groove of Rpp30. Rpp30 is shown in electrostatic surface representation and Rpp29 is shown in cartoon. **e** Cartoon representation of the human Pop5-Rpp14-(Rpp30)₂-Rpp29-Rpp21-Rpp38 heptamer. Bottom left: close-up view of the interface between Rpp30 and Rpp29. Bottom right: close-up view of the interface between Rpp21 and Rpp38

makes extensive interactions with a large area of the RPR S domain, including the T-loops, stem P12 and the extension beyond P12 (Fig. 5c and Supplementary Fig. 12b). L7Ae specifically recognizes the K-turn and helps stem P12.1 fold back onto P12 (Fig. 5c and Supplementary Fig. 12c). Together, two interlinked protein subcomplexes (Pop5-Rpp30)₂ and Rpp29-Rpp21-L7Ae form a single extended protein assembly that presumably helps stabilize the relative positions of the C and S domains of the RPR (Figs 2b, 4c).

**tRNA recognition.** Although pre-tRNA$^{Tyr}$ was used in the cryo-EM analysis of the *Mja*RNase P-tRNA$^{Tyr}$ complex, the ~4.0-Å resolution at the catalytic center suffices to show that the 5′ leader of tRNA is absent in the structure, indicating that the tRNA molecule in the complex is the mature tRNA product after cleavage (Supplementary Fig. 13a). The *Mja*RNase P-tRNA

complex structure reveals that *Mja*RNase P employs a double-anchor mechanism to accommodate the coaxially stacked acceptor arm and TψC arm of tRNA into the substrate-binding pocket (Fig. 6a). In the S domain of RPR, CR-II and CR-III fold into two interleaved T-loops to form one of the anchor (referred to as the T-loop anchor) to stack with the TψC and D loops of tRNA, securing the corner of the L-shaped tRNA in the substrate pocket of *Mja*RNase P (Fig. 6b and Supplementary Fig. 13b). On the other end of the acceptor arm of tRNA, the central A191 (referred to as the A anchor) in the three-nucleotide linker L₅₋₁₅ between stems P5 and P15 of RPR packs on the first base-pair G1-C81 of the acceptor stem of tRNA, anchoring the cleavage site of tRNA right at the catalytic center of *Mja*RNase P (Fig. 6c and Supplementary Fig. 13c). The two RNA anchors respectively locate in the C and S domains of RPR, functioning as a measuring device to recognize the coaxially stacked acceptor and TψC arms of tRNA substrates, which measure a fixed distance of 12 base

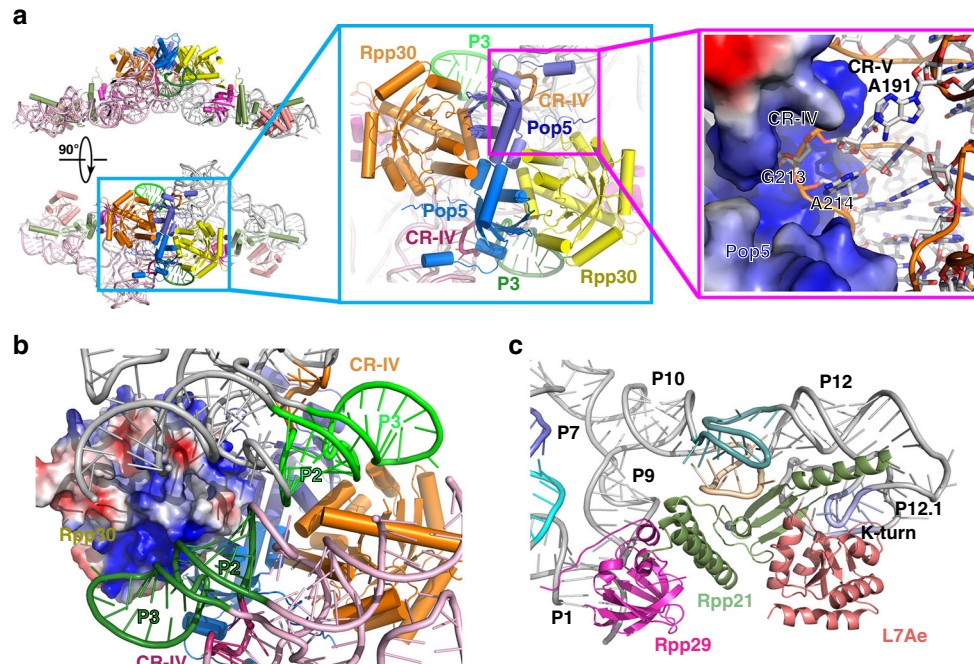

**Fig. 5** Interaction between RPR and RPPs in *Mja*RNase P. **a** Left: The atomic model of the *Mja*RNase P complex is shown in two orthogonal views. RPPs are color-coded as in Fig. 2a. The two RPR molecules are colored in pink and gray, respectively. Middle: Close-up view of the interface between two RPRs (colored in pink and gray) and the (Pop5-Rpp30)$_2$ heterotetramer. The two P3 stems are colored in deepteal and green, respectively. Right: Close-up view of the interface between Pop5 and CR-IV of RPR. Pop5 is shown in electrostatic surface representation and RPR in cartoon representation.
**b** Rpp30 simultaneously interacts with the two RPR molecules in the dimeric holoenzyme. One Rpp30 is shown in electrostatic surface representation and the interacting P2-P3 stem from one RPR is colored in deepteal and the other interacting RPR in green. **c** Close-up view of the interface between the Rpp29-Rpp21-L7Ae heterotrimer and RPR. RPR and RPPs are shown in cartoon and colored as in Fig. 3c and Fig. 2a, respectively

pairs in all tRNA molecules (Fig. 6a)[1,10,48–52]. This double-RNA-anchor for tRNA recognition is conserved in most archaeal and bacterial RPRs[4,23,53,54].

In addition to the two anchors, the 3′-RCCA sequence of bacterial tRNAs is also recognized by a conserved RNA element, loop L$_{15}$, in most bacterial RPRs through base-pairing interactions (Supplementary Fig. 14)[10]. Database search and sequence analysis revealed that some *M. jannaschii* pre-tRNAs do not contain a RCCA sequence at their 3′ termini (Supplementary Fig. 15). This observation is in accordance with the fact that *Mja*RPR lacks a 3′-RCCA recognition element in its small terminal loop at stem P15 (Fig. 3a), suggesting that *Mja*RNase P might have lost this tRNA recognition element during evolution. Indeed, although we used a tRNA with a 3′-RCCA sequence in the *Mja*RNase P-tRNA complex structure, only the first two nucleotides A82 and C83 of the RCCA motif can be modeled into the major groove of the short P15 stem of RPR in the EM density, whereas both C84 and A85 are not visible, presumably disordered in the *Mja*RNase P-tRNA complex structure (Supplementary Fig. 16). Therefore, it is unlikely the 3′-RCCA is recognized by *Mja*RPR through base-pair interactions as in bacterial RNase Ps.

Comparative analysis revealed that *Mja*Pop5 and its eukaryal homologs occupy the same location on their respective RPRs and hold the zigzagged CR-IV of RPR in their deep basic clefts in the same manner (Fig. 6d). In the yeast RNase P-tRNA complex structure, Pop5 stabilizes CR-IV to make direct stacking interactions with nucleobases at the −1, −2, and −3 positions of the 5′ leader of pre-tRNA (Fig. 6d and Supplementary Fig. 17)[40]. The close structural resemblance between *Mja*Pop5 and yeast Pop5 suggests that it is very likely *Mja*Pop5 employs the same mechanism to recognize the 5′ leader of pre-tRNAs (Fig. 6d). In addition to Pop5, other protein components of *Mja*RNase P also make direct contributions to tRNA binding. Rpp30, Rpp29 and

Rpp21 form a continuous highly basic surface that is complementary to the L-shaped tRNA, burying ~890 Å$^2$ interface area between the tRNA and proteins (Fig. 6a). Consistent with this observation, previous biochemical studies showed that a two-residue mutation of *Pho*Rpp29 at this interface substantially reduced the tRNA processing activity of *Pho*RNase P (Supplementary Fig. 18)[36].

**tRNA processing**. The cryo-EM density of the *Mja*RNase P-tRNA complex allowed unambiguous placement of the 5′ end of the mature tRNA in the catalytic center, which resides at the junction between CR-I and CR-V of *Mja*RPR (Fig. 7a and Supplementary Fig. 13a). The spatial arrangement of *Mja*RPR nucleotides around the 5′ end of tRNA, including G40, U41, A233, and A234 as well as the universally conserved uridine U42 in stem P4, highly resembles those observed in *T. maritima* and the newly reported yeast and human RNase P-tRNA complex structures (Fig. 7a)[10,39,40]. In the yeast RNase P structure in complex with a pre-tRNA substrate, equivalent nucleotides A91, U92, U93, G343, and A344 coordinates two catalytic Mg$^{2+}$ ions (Fig. 7a)[40]. Consistent with this structural resemblance, single nucleotide deletion of U42 (ΔU42) or replacement with an adenosine (U42A) greatly diminished the enzymatic activity of *Mja*RNase P, strongly supporting that *Mja*RPR nucleotides at the active site likely play the same role in coordinating two Mg$^{2+}$ ions essential for catalysis (Fig. 7b). Based on the highly conserved architecture at the catalytic center, we propose that the chemical nature of pre-tRNA processing is evolutionarily conserved from bacteria to archaea to eukarya. In this mechanism, an evolutionarily conserved RNA architecture coordinates two Mg$^{2+}$ ions at the catalytic center, one of which (M1) facilitates a hydroxyl ion to perform an S$_N$2-type nucleophile attack at the cleavage site of the pre-tRNA substrate, whereas the other (M2) stabilizes the

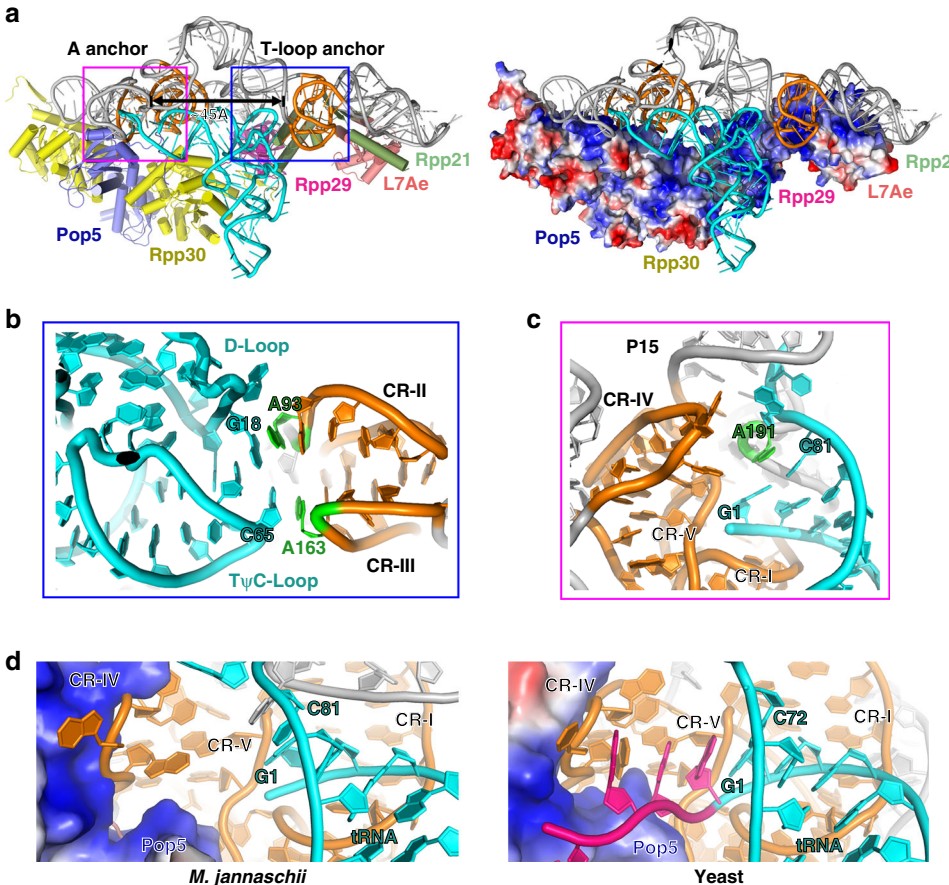

**Fig. 6** tRNA recognition by *Mja*RNase P. **a** Overall structure of the monomeric *Mja*RNase P complex bound with tRNA. Protein components are shown in cartoon (left) and in electrostatic surface (right) representations, respectively. The magenta box denotes the A anchor and the blue box denotes the T-loop anchor. The distance between the two anchors is highlighted with a black double arrow. The tRNA is colored in cyan. **b** Close-up view of the TΨC and D loops of tRNA that stack with the T-loop anchor in CR-II and CR-III of RPR. **c** Close-up view of the central nucleotide A191 (in green) in loop L$_{5-15}$ of RPR that stacks on the first base-pair G1-C81 of tRNA. **d** Left panel: Close-up view of the active site in the *Mja*RNase P-tRNA complex. Right panel: Close-up views of the active site in the yeast RNase P-pre-tRNA complex (PDB: 6AH3). Pop5 is shown in surface representation and colored in electrostatic potential. RPR and tRNA are shown in cartoon and colored in orange and cyan, respectively. The 5′ leader of yeast pre-tRNA$^{Phe}$ is colored in magenta

transition state and mediate proton transfer to the 3′ scissile oxygen (Fig. 7c)[55–63].

**Dimeric organization of *Mja*RNase P**. A surprising observation from the *Mja*RNase P holoenzyme structure is that it adopts a dimeric conformation with a two-fold symmetry (Fig. 2a). The (Pop5-Rpp30)$_2$ heterotetramer sits at the center of the *Mja*RNase P complex and organizes the dimerization (Fig. 5a). Two Pop5 proteins symmetrically recognize two CR-IV regions so that the loops between stems P2 and P4 (nucleotides 223–228) from the two RPR molecules staggered pack together (Fig. 8a). In human and yeast RNase P monomeric structures, the symmetric *Mja*(Pop5-Rpp30)$_2$ heterotetramer is replaced by non-symmetric Pop5-(Rpp30)$_2$-Rpp14 and Pop5-(Rpp30)$_2$-Pop8 heterotetramer respectively, in which only Pop5 is cable of binding CR-IV of the RNAs (Supplementary Fig. 9a)[39,40]. In addition, in both human and yeast RNase P complexes, there is a P19 stem inserted in the loop between stems P2 and P4, which would cause a severe collision if the RNase P complex contained two RNA molecules in a similar manner as in *Mja*RNase P[39,40]. These structural features are conserved in all eukaryal RNase Ps, suggesting that eukaryal RNase Ps should adopt a monomeric but not dimeric configuration.

It is noteworthy that the (Pop5-Rpp30)$_2$ heterotetramer binds to the C domain of both RPR molecules symmetrically in the *Mja*RNase P dimeric complex, so that Rpp30 is involved in tRNA binding in one monomeric complex while sitting on the short P3 stem of RPR from the other complex (Fig. 8b). Markedly, the equivalent Rpp30 that contact the tRNA substrate in both human and yeast monomeric RNase P complexes is also buttressed by additional protein subunit and/or RPR element (Supplementary Fig. 19). Structurally, these interactions appear to stabilize Rpp30 for complex assembly and tRNA substrate binding (Fig. 8b and Supplementary Fig. 19). Consistent with this idea, alanine substitution of *Pho*Rpp30$^{Lys196}$ (equivalent to *Mja*Rpp30$^{Lys198}$) at the interface between Rpp30 and the second RPR in the dimer greatly reduced the pre-tRNA cleavage activity, suggesting that dimeric conformation of archaeal RNase P is crucial for its enzymatic activity (Fig. 8b)[34]. To further examine the function of dimerization, we designed a monomeric mutant *Mja*RPR with an artificial P19 stem inserted between nucleotides G223 and A224, which should preclude the dimer formation (Supplementary Fig. 20). Indeed, in vitro reconstitution with this mutant RPR resulted in a monomeric *Mja*RNase P complex as revealed by both gel filtration and negative staining EM analyses (Fig. 8c, d). In vitro activity assay showed that this monomeric mutant *Mja*RNase P exhibited substantially reduced pre-tRNA processing activity (Fig. 8e), underscoring the importance of dimerization in the in vitro activity of *Mja*RNase P. Whether the *Mja*RNase P holoenzyme adopts a dimeric conformation in vivo is still unclear and warrants further investigations.

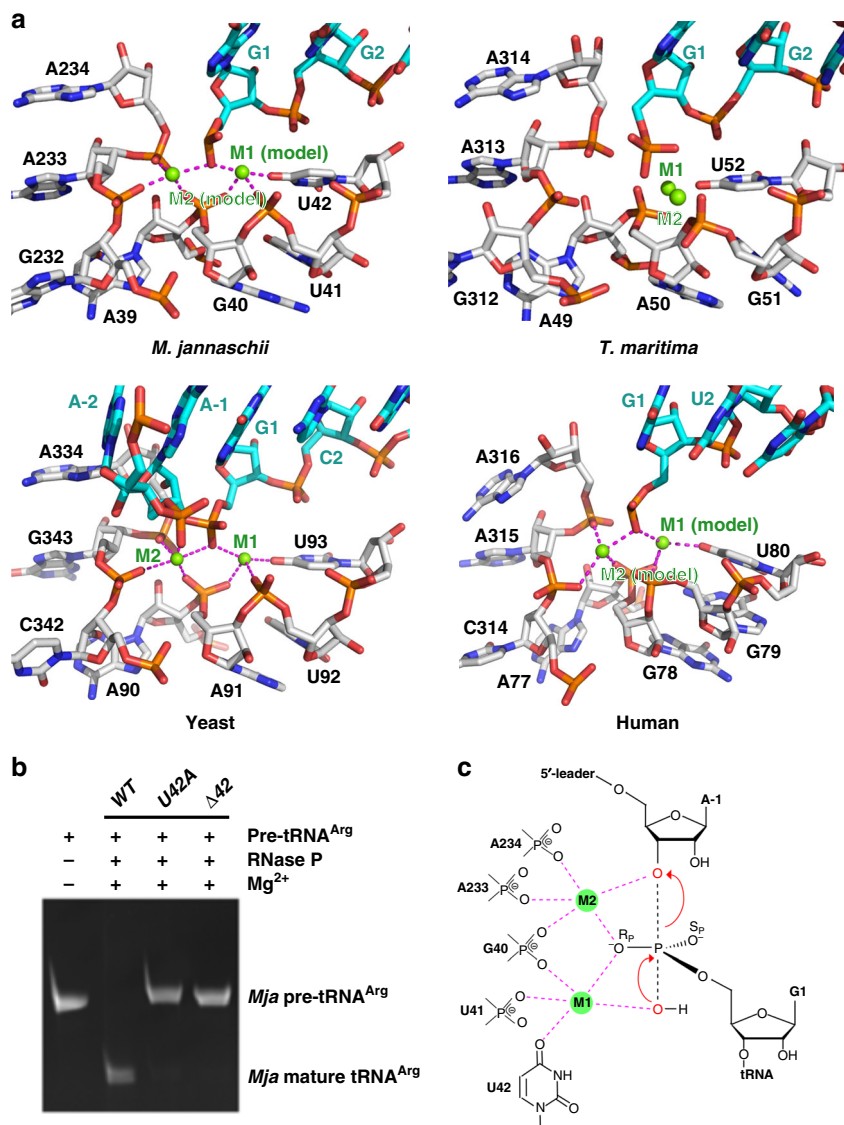

**Fig. 7** The catalytic center of *Mja*RNase P. **a** Close-up views of the catalytic centers of *Mja*RNase P (top left), *T. maritima* RNase P (top right, PDB: 3Q1Q), yeast RNase P (bottom left, PDB: 6AH3), and human RNase P (bottom right, PDB: 6AHU). Two catalytic $Mg^{2+}$ ions (M1 and M2) in the active site of *Mja*RNase P and human RNase P shown in green spheres were modeled based on the yeast RNase P-tRNA complex structure. RPR and tRNA are showed in stick representation and colored in silver and cyan, respectively. The coordination of the two $Mg^{2+}$ ions is denoted by magenta dashed lines. **b** In vitro pre-tRNA processing assay of the *Mja*RNase P holoenzyme reconstituted with WT, U42A and ΔU42 RPR, respectively. **c** Proposed reaction mechanism for 5′-leader cleavage of pre-tRNA by *Mja*RNase P. The reactive oxygens are colored in red, the pre-tRNA scissile phosphate is depicted in a transition state, and the interactions between catalytically important nucleotides and reactive oxygens mediated by $Mg^{2+}$ ions (M1 and M2) are shown as magenta dashed lines

## Discussion

Archaea have been widely used to study the evolution of many biological processes from prokaryotes to eukaryotes. The structure of the *Mja*RNase P holoenzyme reported here provides us a unique opportunity to understand the evolution of RNase P, which accompanied the gradual transformation from an ancient catalytic RNA to a protein-rich RNP.

Although *Mja*RNase P is distinct from bacterial and eukaryal RNase Ps, it contains important structural features from both. *Mja*RNase P employs two RNA anchors, the T-loop anchor and the A anchor in the S and C domains respectively, for tRNA substrate recognition (Figs 6a and 9). This RNA-based apparatus is conserved in the majority bacterial and archaeal RNase Ps[4,23,53,54,64]. In contrast, eukaryal RPRs only maintain the T-loop anchor, and the A anchor is replaced with a protein one

mediated by a eukaryal specific protein Pop1 (Fig. 9)[39,40]. In all cases, the two anchors must be stabilized with a fixed distance that is optimal for accommodating the coaxially stacked acceptor and TψC arms of pre-tRNA substrates (Fig. 6a)[10,39,40]. In bacterial RPR, auxiliary RNA elements mediate long-range RNA-RNA interactions to stabilize the tertiary RNA structure so that the distance between the two anchors is optimal for tRNA binding (Fig. 9 and Supplementary Fig. 6a)[10]. However, *Mja*RPR lacks most of these auxiliary RNA elements (Figs 3a, 9). Consequently, in contrast to bacterial RPRs, the C and the S domains of *Mja*RPR are only loosely connected and require newly evolved protein assembly (Pop5-Rpp30)₂-(Rpp29-Rpp21-L7Ae) to stabilize the RPR for tRNA binding and processing (Figs 6a, 9). This protein-aided stabilization mechanism is faithfully inherited by eukaryal RNase Ps (Fig. 9)[39,40]. Together, these observations

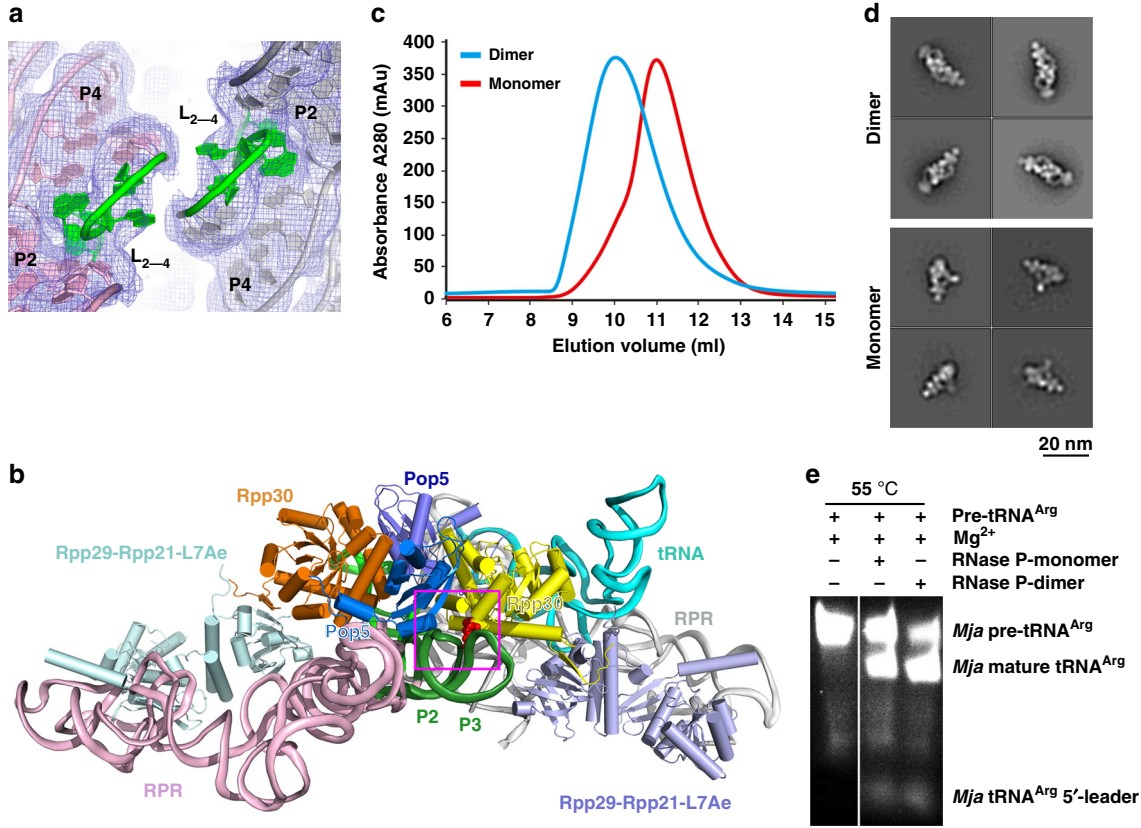

**Fig. 8** Dimeric organization of *Mja*RNase P. **a** The loops between stems P2 and P4 from the two RPRs staggered pack together. The two loops are colored in green and the two RPRs in pink and gray, respectively. The EM density map is shown in light blue mesh. **b** Overall view of the dimeric conformation mediated by the (Pop5-Rpp30)₂ heterotetramer. Rpp30 simultaneously interacts with the two RPR molecules in the dimeric holoenzyme. **c** SEC profiles of the WT (Dimer, blue) and mutant (Monomer, red) *Mja*RNase P complexes. **d** Selected 2D class averages of the WT (Dimer, top) and mutant (Monomer, bottom) *Mja*RNase P complexes. **e** In vitro tRNA processing assay of WT (Dimer) and mutant (Monomer) *Mja*RNase P complexes

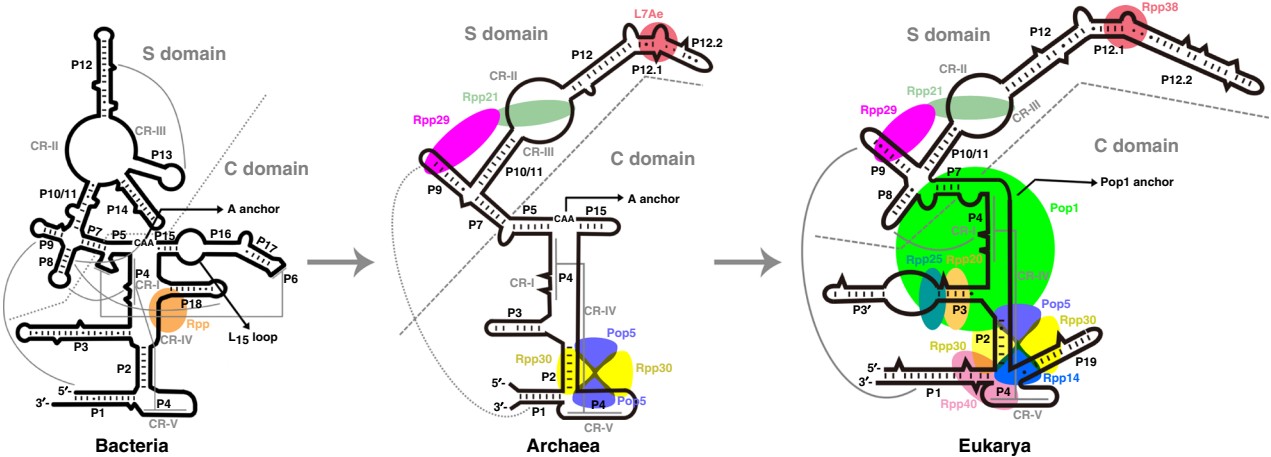

**Fig. 9** A evolution model of RNase P from bacteria to archaea, and to eukarya. Secondary structure models of *T. maritima*, *M. jannaschii* and human RNase P RNAs are based on their atomic structures. Long-range RNA-RNA interactions are denoted by gray lines. Protein components are colored as in Fig. 2a and are placed in positions based on the atomic structures

demonstrate that *Mja*RNase P is a hybrid of bacterial and eukaryal RNase Ps, with bacterial-like two RNA-based anchors but eukaryal-like protein-aided stabilization mechanism, exemplifying that archaea are evolutionary intermediates between bacteria and eukarya.

Bacterial RPR element including loop $L_{15}$ that forms specific base pairs with the 3′-RCCA sequence of tRNA substrates is absent in *Mja*RPR and other M-type RNase Ps (Fig. 3a and Supplementary Figs 6a, 14)[10,22,23,41]. Therefore, similar to yeast and human RNase Ps, the 3′-RCCA of tRNA is not a recognition element in archaeal M-type RNase Ps. It is noteworthy that the 3′-RCCA recognition element is conserved in some A-type archaeal RPRs, and that all *P. horikoshii* tRNA genes contain the 3′ RCCA sequence (Supplementary Fig. 15)[23,54]. It is plausible that *Pho*RNase P might still maintain the 3′-RCCA recognition mechanism and therefore is evolutionarily closer to bacterial

RNase Ps than other archaeal RNase Ps. Structural information of the *Pho*RNase P holoenzyme is needed to test this hypothesis.

P-type archaeal RNase P is a radically minimal form of RNase P[21]. It only contains a C domain but lacks a recognizable S domain[21], suggesting that this type of archaeal RNase P does not have a T-loop anchor as A-type and M-type RNase Ps. In addition, the $L_{5-15}$ loop between stems P5 and P15 in some P-type RPRs only contains two but not three nucleotides[21] so that P-type RNase Ps might also do not have a canonical A anchor. Notably, the complementary region of loop $L_{15}$ that could pair with the 3′-RCCA sequence of tRNA still exists in P-type RPRs[21]. But, some P-type tRNA genes do not contain the 3′-RCCA sequence (Supplementary Fig. 15). Therefore, it is not clear whether P-type RNase P recognizes the tRNA substrate using the 3′-RCCA mechanism. Notwithstanding these dramatic changes, RPRs from P-type RNase Ps are still catalytically active in vitro[21]. The mystery behind P-type archaeal RNase P warrants further mechanistic investigation and structural characterization.

## Methods

**Protein expression and purification.** Full-length genes of *Mja*Pop5 and *Mja*Rpp30 were respectively cloned into the two multiple cloning sites of the pETDuet vector with a 6 × His tag at the N-terminus of *Mja*Rpp30 and were expressed in *E. coli* BL21 (DE3) cells. After induction for 18 h with 0.1 mM IPTG at 18 °C, the cells were harvested and the pellets were resuspended in buffer A (50 mM Tris-HCl, pH 8.0, 1 M NaCl and 0.8 M Urea). The cells were then lysed by sonication and the cell debris was removed by centrifugation. The supernatant was mixed with Ni-NTA agarose beads (Qiagen) and rocked for 4 h at 4 °C before elution with buffer B (50 mM Tris-HCl, pH 8.0, 1 M NaCl and 500 mM imidazole). The protein elution was heated at 65 °C for 30 min and the contaminants were removed by ultracentrifugation. The proteins were further purified by ion-exchange and SEC. The purified proteins in buffer C (50 mM Hepes, pH 7.5, 50 mM MgCl$_2$, 200 mM NaCl, 500 mM KCl, 50 mM potassium acetate) were concentrated to ~20 mg mL$^{-1}$ and stored at −80 °C. The same procedure was used to express and purify *Mja*L7Ae and the *Mja*Rpp29-*Mja*Rpp21 heterodimer.

**RNA oligonucleotides.** The RPR was generated by in vitro run-off transcription with T7 RNA polymerase at 37 °C. During the transcription, the purified *Mja*L7Ae was added as a chaperone to help stabilize the newly transcribed *Mja*RPR. The RPR was ultracentrifuged to remove the precipitants and then further purified by SEC with a Superdex200 column (GE Life Science) equilibrated with buffer C and was concentrate to ~10 mg mL$^{-1}$. The *Mja*RPR mutants were transcribed and purified similarly. The *E. coli* pre-tRNA$^{Tyr}$, *M. jannaschii* pre-tRNA$^{Arg}$, and human pre-tRNA$^{Val}$ were prepared as described above.

**MjaRNase P holoenzyme complex assembly.** Biochemical reconstitution of the *Mja*RNase P holoenzyme was performed by incubating *Mja*RPR with the protein components with a molar ratio of 1:3 in buffer C at 37 °C for 30 min, and then at 55 °C for another 30 min. The *Mja*RNase P holoenzyme was then purified by SEC with a Superdex200 column (GE Life Science) equilibrated with buffer C. The purified *Mja*RNase P holoenzyme was concentrate to ~5 mg mL$^{-1}$. *Mja*RNase P was mixed with *E. coli* pre-tRNA$^{Tyr}$ in a molar ratio of 1:10 to generate the *Mja*RNase P-pre-tRNA complex sample for EM experiments.

**pre-tRNA processing assays.** To characterize the activity of the reconstituted WT and mutant *Mja*RNase P holoenzyme, pre-tRNA substrate was mixed with *Mja*RNas P holoenzyme with a molar ratio of 100:1 in 10 µL buffer C at 25 °C or 55 °C for 30 min. The reactions were quenched by adding the loading dye. The samples were then loaded into a 15% urea denaturing polyacrylamide gel with TBE buffer and stained with ethidium bromide.

**Electron microscopy.** The reconstituted *Mja*RNase P complex was first studied by the negative-staining EM. Copper grid coated with a thin carbon film was glow-discharged for 20 s and 5 µL sample was applied onto it. The sample was incubated for 1 min and stained with 1% (w/v) uranyl formate for another minute. About 300 micrographs were taken in Tecnai G2 Spirit microscope operated at 120 kV and recorded on a 4k × 4k CCD camera with a defocus of ~−2 µm and a magnification of 67,000 × (1.74 Å pixel$^{-1}$).

Cryo-EM samples were prepared with Vitrobot Mark IV (FEI) at 4 °C and 100% humidity. Ted Pella lacey copper grids coated with a thin layer of continuous carbon film were glow–discharged for 15 s and 4 µL of the sample was applied (blotting parameters: waiting time 20 s, blot force −1, blot time 4 s) and plunge-frozen in liquid ethane. Since the sample was in a high-salt buffer, which hinders the cryo-EM data acquisition, the high-salt buffer was exchanged to a buffer

contains 200 mM salt (40 mM Hepes, pH 7.5, 200 mM NaCl) on the grids quickly before plunge. For the *Mja*RNase P-tRNA complex, cryo-EM data was acquired on a Titan Krios (FEI) operated at 300 keV equipped with a K2 Summit direct detector. The detector was operated in the super-resolution mode at 120,000 × magnification with a calibrated physical pixel size of 1.32 Å. Data acquisition was carried out with SerialEM[65] to record 4144 movies with a defocus range from −1.5 to −2.5 µm and with a total exposure time of 8 s. 32 frames were collected per movie at a dose rate of 6.2 e$^-$ Å$^{-2}$ s$^{-1}$, which provided a total accumulated dose of 50 e$^-$ Å$^{-2}$. For the *Mja*RNase P holoenzyme, data were collected using the Folcan III camera in counting mode at a magnification of 81,000 × (1.09 Å pixel$^{-1}$). 2611 movies with defocus values ranging from −1.5 to −2.5 µm were acquired using EPU software with a total exposure time of 69 s. 32 frames were collected per movie with a total accumulated dose of 40 e$^-$ Å$^{-2}$.

**Image processing.** For negative staining analysis, ~10,000 particles were picked using e2box.py from EMAN2[66]. Then all the subsequent steps of extraction, classification and initial model building were performed in RELION 2.1[67]. The initial model was used for 3D refinement, which result in a map with a 23-Å resolution. The map of the negative-stained *Mja*RNase P was low-pass filtered to 60 Å and used for cryo-EM data processing.

For cryo-EM analysis of the *Mja*RNase P-tRNA complex dataset, we applied MotionCor2[68] to perform the frame alignment and dose-weighting, and Gctf[69] to estimate the contrast transfer function (CTF) parameters. An initial set of approximately 2,000 particles were manually picked, then 2D class averages were calculated and used as reference for automatic picking with Gautomatch (http://www.mrc-lmb.cam.ac.uk/kzhang/Gautomatch/). About 930,000 particles were auto-picked and selected for further 2D classification, which yielded a dataset containing about 829,000 particles. Then all particles were subjected to 3D refinement using a 60 Å low pass filtered negative-stained map as a reference. The refined model was used as reference for 3D classification. The major class with reasonable features containing about 15,0000 particles was applied for further 3D refinement. Since *Mja*RNase P appears as a dimeric complex, the C2 symmetry was applied during 3D refinement, producing a final map at a 4.3-Å resolution based on the gold-standard FSC cut-off criterion at 0.143. The density map was sharpened by applying a negative temperature factor automatically estimated by post-processing program of RELION 2.1[67]. Local resolution estimates were determined using RELION 2.1[67]. For cryo-EM analysis of the apo *Mja*RNase P dataset, image processing procedures were the same as described above.

**Model building.** De novo atomic model building and rigid-docking of homologous structures are combined to build the model of the entire *Mja*RNase P complex with or without tRNA. Model building of the C domain of *Mja*RPR was mostly based on the bacterial RPR structure[9,10], whereas the S domain mostly based on the human RPR structure[39]. We first modeled the characteristic three coaxially stacked RNA stems with standard double-stranded RNAs based on the bacterial RNase P structure (PDB: 3Q1Q). Then the conserved structure of CR-I-II-III-IV-V regions were identified and the models were fit into the EM density according to the bacterial and human RNase P, respectively (PDB: 3Q1Q and 6AHU). Next we built the K-turn by fitting the *Pho* P12.1 and P12.2 (PDB: 5DCV; 5XTM; 5Y7M) into the EM density. After assignment of all these elements, junction regions and single-stranded loops were then built manually with stereochemistry considerations. RNA nucleotide register was judged by conserved and unique regions of RPR, including the CR-I-IV-V pseudo-knot, CR-II-III T-loop anchor, the A-anchor and the K-turn. The final RPR structure satisfies the requirement of all these special regions as well as all the helical regions. The crystal structure of *E. coli* tRNA$^{Tyr}$ was determined previously (PDB: 4V8D)[70]. We fit this crystal structure into the EM density with minor adjustment.

As for RPPs, we built their models based on the crystal structures of *M. jannaschii* and *P. horikoshii* RNase P protein subunits[34,36,71]. The crystal structure of *Mja*L7Ae has already been determined (PDB: 1SDS)[71]. We directly fit *Mja*L7Ae structure into the EM density with minor adjustment. For other protein components, we first fit each *Pho* proteins (PDB: 3WZ0; 2ZAE) into the EM density by rigid-body docking followed by manual adjustment and de novo model building for regions that are different from *Pho* protein structures. There are four major differences between *Mja* and *Pho* protein structures. First, *Mja*Rpp30 has an extra β strand docking on *Mja*Rpp29. Second, the C-terminal tail of *Mja*Rpp29 sticks out to fit into a hydrophobic groove of *Mja*Rpp30 whereas the C-terminal tail is missing in the *Pho*Rpp29 structure. There are clear electron densities corresponding to these extra regions in the EM density map (Supplementary Fig. 10a, b). It is these extra structural elements of *Mja*Rpp30 and *Mja*Rpp29 that mediates the connection between the *Mja* (Pop5-Rpp30)$_2$ heterotetramer and the *Mja* Rpp29-Rpp21-L7Ae heterotrimer (Fig. 4c). The third difference is that *Mja*Rpp21 contains extra C-terminal residues that fold into a helix packing on both *Mja*L7Ae and the junction between stems P12 and P12.1/P12.2 of *Mja*RPR (Fig. 5c and Supplementary Figs 5, 9b, 12a). Another difference is from *Mja*Pop5, which has a short extra helix packing on the small terminal loop of stem P15 of *Mja*RPR (Fig. 4c and Supplementary Figs 5, 9b). The EM density clearly shows the extra density of this helix (Supplementary Figs 5, 12a).

**Reporting summary**. Further information on research design is available in the Nature Research Reporting Summary linked to this article.

## Data availability

The data that support the findings of this study are available from the corresponding author upon reasonable request. The cryo-EM 3D maps of the *Mja*RNase P holoenzyme and the *Mja*RNase P-tRNA complex were deposited in EMDB database with accession codes EMD-9899 and EMD-9900, respectively. The atomic models were deposited in PDB with accession codes 6K0A and 6K0B, respectively.

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

## Acknowledgements

We thank the staff members of the Electron Microscopy System, specially, M. Cao, for his help on data collection. And we also thank the Mass Spectrometry System at Shanghai Institute of Precision Medicine for providing technical support and assistance in data collection. This work was supported by grants from the National Natural Science Foundation of China (31525007 to M.L.), the Strategic Priority Research Program of the Chinese Academy of Sciences (XDB08010201 to M.L.) and the Young Elite Scientist Sponsorship Program of China Association for Science and Technology (2018QNRC001 to P.L.).

## Author contributions

F.W., J.T., J.C., and S.S. reconstituted the *Mja*RNase P complex. Q.W. and M.T. prepared cryo-EM specimens, collected datasets and determined the structures. J.W. carried out model building and refinement. All the authors were involved in data interpretation and contributed the writing of the manuscript. M.L., Q.W., and P.L. wrote the manuscript. M.L., J.W., and P.L. initiated and orchestrated the project.

## Additional information

**Competing interests:** The authors declare no competing interests.

**Journal Peer Review Information** *Nature Communications* thanks the anonymous reviewer(s) for their contribution to the peer review of this work. Peer reviewer reports are available.

