## [Peer Review File · Nature Communications]

Reviewers' comments:

Reviewer #1 (Remarks to the Author):

The authors present a detailed model of the "type M" RNase P holoenzyme from the archaeon *M. jannascii*, bound to tRNA, the product of the cleavage reaction mediated by the enzyme. By recombinantly expressing the enzyme's five protein subunits and combining with the in vitro transcribed RNA subunit (RPR) and adding a tRNA substrate, the authors were able to generate particles with sufficient homogeneity and resolution to build a map with an average resolution calculated to be ~4.7 Å. To build the atomic level model, the authors combined both rigid body docking of atomic structures available in the PDB, with de novo model building. The authors find the enzyme to form a dimer and impose C2 symmetry during refinement. The resulting structure provides unparalleled insights into the structure and assembly of this ancient and essential protein-aided ribozyme, and the findings will immensely impact efforts to understand the structure and function of the homologous enzymes from eukaryotic organisms.

Despite its very high significance, the manuscript suffers from deficiencies that make it unsuitable for publication in its current form:

1. The authors do not adequately describe how their model was built. A description is provided in lines 383-393, and generally states that available crystal structures of homologs of the proteins were docked into the density, then adjusted manually; for the RNA, the authors used homology to the bacterial RPR secondary structure, while junctions between secondary structural units were built manually. The problem is that the resulting structures differ in important ways from the input homology models, and the new features, and evidence supporting them is not adequately provided. E.g., Rpp30 is modeled with an extra beta strand that docks on RPP29; Rpp21 is modeled with an extra helix that contacts Rpp38. In addition, there are several sections in which individual RNA nucleotides are described making specific interactions with the proteins, or tRNA. This major weakness makes the validity of many of the authors' structure-based conclusions difficult to evaluate.
2. The authors do not adequately distinguish what are the new findings of the current work from those that support findings published previously (by others). Glaring examples are: (i) line 226: the authors propose a two-metal mechanism for RNA cleavage by RNase P, citing classical paper of Steitz & Steitz (1993), but seem to ignore that others have proposed such a mechanism for RNase P. (ii) Line 105 and extended data fig. 5, the authors state that k-turns are a conserved feature of archaeal RNase Ps; several recent papers in the literature have made this assertion, and this is not acknowledged here, nor do authors provide adequate evidence to support the assertion on the basis of their findings.
3. Lastly, the authors frequently attribute functional roles to structural features, without supporting functional evidence. For example, they often state that particular features "stabilize" some second feature of the structure; such an assertion would require evidence that without the first element, the second is not formed (e.g., line 136 – what is the evidence that the k-turn is not formed in the absence of Rpp38?; line 153; what is the evidence that the RPR needs to be stabilized by the proteins?). The authors should either provide evidence, or clarify that their functional attribution is speculation based on the model.

Additional comments:

4. There is generally insufficient discussion of how the structure and its components differ from the input models, including proteins, RPR (*Mja* v. *Pho*) and tRNA (*Tyr* v. *Phe*)

5. line82: The analogy to the flying pterosaur is cute, but given the C2 symmetry of the structure, misleading.
6. Page 5, lines 98 &103: there is no extended data fig. 4c
7. Page 5, line 111: The descriptor of the proteins acting as a "extended hinge" is unclear and/or not supported by the data
8. Page 6, line 118: It is unclear whether the data provides the resolution needed to see the mentioned base and side-chain interactions. The potential importance of the interaction is not supported by evidence.
9. Page 7, line 147, and figure 3b; it looks like the labels on the tRNA arms are swapped.
10. line 186, figure 4, the authors don't justify inclusion of type A archaeal RNase P
11. line 204; functional equivalence of type M and type P RPR is unclear and/or not well justified
12. line 211-213; the authors present no evidence supporting structural or evolutionary assertions about CCA recognition by eukaryal RNase P.
13. Figure 2 legend; the authors might consider "colored to match label", instead of describing colors such as "slate", "forest" and "deeptale".
14. Figure 2b, line 270; the dotted line shows the conserved, and well-known P4 pseudoknot, not tertiary contacts observed in the structure.
15. Figure 3a; the star and triangle used to mark the recognition features are hard to see
16. Figure 4 and extended figure 7 are not addressed in the body of the article.
17. "Tailor" when meaning "tail"?
18. The authors state that the active site geometry is "ideal" for binding two magnesium ions; it is not clear whether their data adequately constrain the model to make significant new conclusions.

Reviewer #2 (Remarks to the Author):

RNase P is an endoribonuclease involved in the processing of RNA transcripts including precursor tRNA, other non-coding RNA precursors and mRNA. RNase P consists of protein and RNA and it is the RNA that is responsible for the catalytic activity. It has been shown to be present in all kingdoms of life and based on the secondary structure RNase P RNA (RPR) is classified into different types: Type A, Type B, Type M, Type P and Type "Eukarya", where A, M and P are found among the archaea. Wan et al. present the Cryo-EM structure of a reconstituted archaeal RNase P-tRNA complex at 4.3Å resolution. The complex was reconstituted using in vitro transcribed *Methanocaldococcus jannaschii* RPR, a Type M RPR, and recombinant RNase P proteins. They provide data demonstrating that the reconstituted RNase P cleaves a precursor tRNA. Their findings are interesting and provide insight into the structure of an archaeal RNase P. However, I have major concerns and comments that needs to be addressed and might help to improve the manuscript.

Issues related to the activity. In order to evaluate the findings, the following should be included:

1. Information about the precursor tRNA that was used, a figure showing its secondary structure would be informative. The best would be if the authors used a *M. jannaschii* pre-tRNA.
2. In Fig 1b the band representing the 5' matured tRNA is shown while the 5' leader is not seen. Please also show the mobility of the 5' leader. This is relevant in order to understand that the reconstituted RNase P cleave at the correct site. In this context, you need to include controls and one control is how the reconstituted RNase P perform in relation to native RNase P. Also, it is informative to follow the reaction over time both for the reconstituted RNase P and native RNase P.
3. What is the final Mg²⁺ concentration?
4. Do you have any information about the composition of the native RNase P? Does it appear as dimer in the cell?

For the discussion about the function/ mechanism of RNase P cleavage I recommend that you refer to previous reports. Much of the discussion is not new since it has been discussed/ reported elsewhere. For example, the discussion about the interaction between the tRNA T-/D-loop region and RNase P, cleavage mechanism (involvement of two metal ion), interactions between the substrate and the RPR and the role of proteins, measuring mechanism etc. For references, see e.g. Liu F, Altman S. 2010. Ribonuclease P. Protein Reviews vol 10. Springer, New York Dordrecht Heidelberg, London, and chapters and references therein. After 2010 there has been reports relevant to the present study and I strongly recommend to go through these and take these publications into consideration.

Since the 5' leader is not detected in the complex the structure represents the post-cleavage state and the data should be discussed in this perspective.

The Type M lacks residues in P15-loop that can pair with the pre-tRNA 3' end (3' NCC-motif) and the bound tRNA has a 3' CCA, but information whether the tRNA genes in *Methanocaldococcus jannaschii* encodes the 3' CCA needs to be included (see also my comment about the origin of the pre-tRNA above). If the tRNA genes do not encode CCA might provide a rational to why residues that can pair with the pre-tRNA 3' end is absent.

When you discuss the P type RPR I recommend that you also discuss the catalytic performance of the C-domain lacking the S-domain.

In the text please define "...the T-loops..." (e.g. line 135) so the reader clearly understand that you are not discussing the tRNA T-loop.

Line 29 "...one of only two ribozymes..", which is the other ribozyme?

Lines 54-57 The sentence "Type A RNA that largely..." is misleading please revise.

Line 189-192 Do you have any biochemical evidence to support this statement? Please clarify.

Line 199-200 There are exceptions to this in Bacteria, e.g., *Chlamydia* spp. and certain cyanobacteria.

Line 209-220 This is not the case since RNase P processing is affected when the interaction with the

pre-tRNA 3' end and RNase P is disturbed. But this is dependent on the pre-tRNA structure.

Line 312-313 How did you removed the protein contaminations after heat treatment?

Line 320 What was the rational to add Rpp38? Please clarify.

Line 326 Please define KoAc?

Extended data

Extended Data Figure 3 legend. Please included references for the crystal structures of *P. horikoshii* proteins.

Extended Data Figure 6 legend. Please expand the text after (d), e.g., how do you define ideal?

Extended Data Figure 8 legend. *T. maritime* should be *T. maritima* (also in Extended Data Figure 9 legend). Also, give reference Reiter et al. 2010.

Extended Data Figure 10 legend. Include reference 16 in the figure legend text. Change *Pyrococcus horikoshii* to *P. horikoshii*.

Reviewer #3 (Remarks to the Author):

In this manuscript Wan et al. present the 4.3Å cryo-EM structure of an M-type archeal RNase P-tRNA complex, an important enzyme class that removes the 5' leader from pre-tRNA during tRNA biogenesis. Owing to the availability of structural homology of the RNase P proteins and RNA, the authors could produce a near-atomic model of the enzyme bound to a product tRNA. Based on the structure, the authors describe three main findings. First, they propose a 'double-anchor' mechanism in which the co-axially stacked T and Acceptor stem loop of the tRNA are used to discriminate non-tRNA substrates. Second, the authors observe tRNA CCA tailing sequence is not recognized with sequence-specificity by M-type RNase P enzymes. Third, the authors compare the M-type archeal RNase P to its alternative archeal, bacterial, and eukaryotic variants. Taken together, the manuscript makes a valuable addition to our understanding of the function and evolution of RNase P enzymes. Pending a few changes and clarifications, I would happy to support the paper for publication in Nature Communications.

Major comments

1. The authors suggest that the tRNA 3' CCA is not recognized with sequence-specificity. This would be an important conclusion of the study and should thus be supported by the structural data. The authors should show the density for the 3'CCA and its RPR interaction. For example, if the path of the RNA backbone were ambiguous, the CCA bases might instead interact with the RPR through triplet-helix interactions. At 4.3Å such details may be difficult to observe for nucleic acid in particular, depending on the local resolution.
2. Related to point 1, while sequence-specific recognition of the CCA element may not be required, its presence may be important for cooperative binding of the tRNA in addition to the T and Acceptor loop interaction. Do the authors have data on how sequence changes or absence of the CCA element affect the in vitro binding affinity/ cleavage activity of pre-tRNA to RNaseP? If there is no clear data on this, the authors should reflect this in their model (text and Fig 4) that the presence of the CCA may still be needed for pre-tRNA specificity, even if the sequence is not read out directly. This should also be

reflected in the Abstract.

3. Can the authors speculate how RNase P dimerization might influence cleavage activity, as this seems to be a conserved feature?
4. The Abstract is misleading to the nature of the 3'CCA recognition (Line 44) in this structure. The authors should rephrase this to clarify that the 3'CCA element recognition may not be important for M-type RNase P enzymes.

Minor comments

1. It is difficult to follow the RNA elements between figures 2a/b and 3 due to the 180 degree rotation of the structure. If the authors wish to keep this arrangement, please improve the labeling of Figure 3a to include: tRNA acceptor and TyC arms, modules A/B, tRNA 5' and 3' ends, and the RPR elements.
2. The authors should correct the text for typos and use of grammar, as I spotted a few errors throughout.
3. Please mention in the main text that C2 symmetry was used to improve the resolution, even though it is already stated in the methods.
4. Line 30: Please correct to "only one of two ribozymes present"
5. Line 72: It would be useful to mention here that the pre-tRNA was cleaved during sample preparation, since this otherwise might lead to confusion with the line above (Line 70: "[...] MjRNase P and a pre-tRNA^{tyr} substrate at a ratio of 1:10 [...]").
6. Line 145: Please repeat here that pre-tRNA^{tyr} is being used in the co-complex. Related to this, is the distance between the T and Acceptor stems invariant in all tRNAs, and is this worth noting in the text?
7. For Figures 3a, c, d, please indicate the site of cleavage the 5' end of the tRNA or indicate this in the Figure legend.
8. In Figure 4, please shorten the RPR length in the P-type RNase P model by 2/3 to more accurately represent its unique architecture.
9. Do the authors find a tRNA-free population of RNase P in their 3D classification, and if so, does this yield any insights into the tRNA binding mechanism?
10. To better judge the density quality, it would be helpful to include fits of the density with the proteins and RNAs in the supplement. Zoom-ins of important regions would be helpful too.

Point-to-point responses to reviewers' comments

Cryo-EM structure of an archaeal ribonuclease P holoenzyme in complex with tRNA

Reviewer 1

The resulting structure provides unparalleled insights into the structure and assembly of this ancient and essential protein-aided ribozyme, and the findings will immensely impact efforts to understand the structure and function of the homologous enzymes from eukaryotic organisms.

Thanks!

1. The authors do not adequately describe how their model was built. A description is provided in lines 383-393, and generally states that available crystal structures of homologs of the proteins were docked into the density, then adjusted manually; for the RNA, the authors used homology to the bacterial RPR secondary structure, while junctions between secondary structural units were built manually. The problem is that the resulting structures differ in important ways from the input homology models, and the new features, and evidence supporting them is not adequately provided. E.g., Rpp30 is modeled with an extra beta strand that docks on RPP29; Rpp21 is modeled with an extra helix that contacts Rpp38. In addition, there are several sections in which individual RNA nucleotides are described making specific interactions with the proteins, or tRNA. This major weakness makes the validity of many of the authors' structure-based conclusions difficult to evaluate.

Recently, we determined the cryo-EM structures of both human (3.7 Å) and yeast (3.5Å) RNase P complexes, providing two additional RPR structures [1] [2]. Model building of the C domain of *Mja*RPR was mostly based on the bacterial RPR structure [3, 4], whereas the S domain mostly based on the human RPR structure [1]. We first modeled the characteristic three coaxially stacked RNA stems with standard double-strand RNAs based on the bacterial RNase P structure (PDB: 3Q1Q). Then the conserved structure of CR-I-II-III-IV-V regions were identified and the models were fit into the EM density according to the bacterial and human RNase P, respectively (PDB: 3Q1Q and 6AHU). Next we built the K-turn by fitting the *Pho* P12.1 and P12.2 (PDB: 5DCV; 5XTM; 5Y7M) into the EM density[5, 6]. After assignment of all these elements, junction regions and single-stranded loops were then built manually with stereochemistry considerations. RNA nucleotide register was judged by conserved and unique regions of RPR, including the CR-I-IV-V pseudo-knot, CR-II-III T-loop anchor, the A-anchor and the K-turn. The final RPR structure satisfies the requirement of all these special regions as well as all the helical regions.

As for the proteins, we built their models based on the crystal structures of *M. jannaschii* and *Pyrococcus horikoshii* RNase P protein subunits [7-9]. The crystal structure of *Mja* L7Ae has already been determined (PDB: 1SDS) [7]. So we directly fit its structure into the EM density with minor adjustment. For other protein components, we first fit each *Pho* protein (PDB: 3WZ0; 2ZAE) into our EM density

by rigid-body docking followed by manual adjustment and *de novo* model building for regions that are different from *Pho* protein structures. There are four major differences between *Mja* and *Pho* protein structures. First, *MjaRpp30* has an extra β strand docking on *MjaRpp29*. Second, the C-terminal tail of *MjaRpp29* sticks out to fit into a hydrophobic groove of *MjaRpp30* whereas the C-terminal tail is missing in the *PhoRpp29* structure. There are clear electron densities corresponding to these extra regions in the EM density map (Supplementary Fig. 10a-b). It is these extra structural elements of *MjaRpp30* and *MjaRpp29* that mediates the connection between the *Mja* (Pop5-Rpp30)₂ heterotetramer and the *Mja* Rpp29-Rpp21-L7Ae heterotrimer (Fig. 4c). Notably, except for a few A-type archaeal RNase Ps, the C-terminal tail of Rpp30 is highly conserved in most archaeal Rpp30 proteins, suggesting that the intermolecular connection between Rpp30 and Rpp29, and therefore, the overall architecture of the protein assembly is very likely conserved in most archaeal RNase P holoenzymes (Supplementary Fig. 11). In addition, the inter-subcomplex (Pop5-Rpp30)₂—(Rpp29-Rpp21-L7Ae) connection mediated by Rpp30 and Rpp29 is also conserved in human RNase P holoenzyme structure, suggesting that this structural feature is likely evolutionarily conserved from archaea to eukarya (Fig. 4c and Fig. 4e) [1]. The third difference between *Mja* and *Pho* protein structures is that *MjaRpp21* contains extra C-terminal residues that fold into a helix packing on both *MjaL7Ae* and the junction between stems P12 and P12.1/P12.2 of *MjaRPR* (Fig. 5c and Supplementary Figs. 5, 9b and 12a). Another difference is from *MjaPop5*, which has a short extra helix packing on the small terminal loop of stem P15 of *MjaRPR* (Fig. 4c and Supplementary Figs. 5 and 9b). The EM density clearly shows the extra density of this helix (Supplementary Figs. 5 and 12a).

The crystal structure of *E. coli* tRNA^{Tyr} was determined previously (PDB: 4V8D) [10]. We fit this crystal structure into the EM density with minor adjustment.

We have included the detailed description of model building in the method section in the revised manuscript.

In addition, there are several sections in which individual RNA nucleotides are described making specific interactions with the proteins, or tRNA. This major weakness makes the validity of many of the authors' structure-based conclusions difficult to evaluate.

In the original manuscript, we mentioned specific RNA interactions in three places.

- (1) Line 117-118 in original manuscript, "In particular, the nucleobase of G213 is sandwiched between Ile104 and Trp120 of Pop5, securing CR-IV in the basic cleft (Fig. 2f and Extended Data Fig. 6a)."

We agree with the reviewer that the resolution of the EM density map cannot provide unambiguous details to support the side-chain interactions between G213 of RPR and Ile104 and Trp120 of Pop5. Therefore, we removed this sentence in the revised manuscript.

- (2) Line 145-148 in original manuscript, "Two nucleotides A93 and A163 from the T-loops in the CR-II/CR-III motif of RPR serve as an anchor (the T-loop-anchor), respectively stacking on the bases of G18 and C65 in the T ψ C and D loops at the

corner of the “L”-shaped tRNA.”

We modified this section as “In the S domain of RPR, CR-II and CR-III fold into two interleaved T-loops to form one of the anchor (referred to as the ‘T-loop anchor’) to stack with the T ψ C and D loops of tRNA, securing the corner of the ‘L’-shaped tRNA in the substrate pocket of *Mja*RNase P” (Fig. 6b and Supplementary Fig. 13b). This CR-II/CR-III mediated interaction with the T ψ C and D loops of tRNA is conserved in bacterial and eukaryotic RNase P structures [1, 4]. Although individual nucleotides of RPR and tRNA cannot be distinguished in the EM density map, both CR-II/III of RPR and the T ψ C and D loops of tRNA can be nicely fit into the EM density (Supplementary Fig. 13b).

- (3) Line 148-152, “At the terminus of the acceptor arm of tRNA, the central nucleotide A191 in linker L₅₋₁₅ between stems P5 and P15 of RPR (hereafter referred to as the A-anchor) packs on the first base-pair G1-C81 of the acceptor stem, anchoring the cleavage site of tRNA right at the catalytic center of the ribozyme.”

We modified this section as “On the other end of the acceptor arm of tRNA, the central A191 (referred to as the ‘A anchor’) in the three-nucleotide linker L₅₋₁₅ between stems P5 and P15 of RPR packs on the first base-pair G1-C81 of the acceptor stem of tRNA, anchoring the cleavage site of tRNA right at the catalytic center of *Mja*RNase P (Fig. 6c and Supplementary Fig. 13c).” Similar to the T-loop anchor, the A-anchor is conserved in most bacterial RNase Ps [11]. Supplementary Fig. 13c shows the stereo view of the EM density map at the A-anchor. Again, although individual nucleotides cannot be distinguished, A191 of RPR and the G1-C81 base pair of tRNA can be nicely fit into the density with the identical conformation as in *T.maritima* RNase P-tRNA structure (Supplementary Fig. 13c) [4]. Therefore, we conclude that *Mja*RNase P very likely employs the same ‘two-RNA anchor’ mechanism for tRNA recognition.

2. The authors do not adequately distinguish what are the new findings of the current work from those that support findings published previously (by others).

- (1) line 226: the authors propose a two-metal mechanism for RNA cleavage by RNase P, citing classical paper of Steitz & Steitz (1993), but seem to ignore that others have proposed such a mechanism for RNase P.

Thanks for this good point. We now cite several previous papers that have proposed the two-metal ion mechanism for RNase P activity in the revised manuscript [12-20]. (Page 15).

12. Steitz, T.A. and J.A. Steitz, A general two-metal-ion mechanism for catalytic RNA. Proc Natl Acad Sci U S A, 1993. 90(14): p. 6498-502.
13. Liu, X., Y. Chen, and C.A. Fierke, Inner-Sphere Coordination of Divalent Metal Ion with Nucleobase in Catalytic RNA. J Am Chem Soc, 2017. 139(48):

- p. 17457-17463.
14. Kirsebom, L.A. and S. Trobro, RNase P RNA-mediated cleavage. *IUBMB Life*, 2009. 61(3): p. 189-200.
 15. Perreault, J.P. and S. Altman, Important 2'-hydroxyl groups in model substrates for M1 RNA, the catalytic RNA subunit of RNase P from *Escherichia coli*. *J Mol Biol*, 1992. 226(2): p. 399-409.
 16. Fedor, M.J., The role of metal ions in RNA catalysis. *Curr Opin Struct Biol*, 2002. 12(3): p. 289-95.
 17. Hsieh, J., et al., A divalent cation stabilizes the active conformation of the *B. subtilis* RNase P x pre-tRNA complex: a role for an inner-sphere metal ion in RNase P. *J Mol Biol*, 2010. 400(1): p. 38-51.
 18. Christian, E.L., et al., The P4 metal binding site in RNase P RNA affects active site metal affinity through substrate positioning. *RNA*, 2006. 12(8): p. 1463-7.
 19. Beebe, J.A., J.C. Kurz, and C.A. Fierke, Magnesium Ions Are Required by *Bacillus subtilis* Ribonuclease P RNA for both Binding and Cleaving Precursor tRNA^{Asp}. *Biochemistry*, 1996. 35(32): p. 10493-10505.
 20. Scott, W.G. and A. Klug, Ribozymes: structure and mechanism in RNA catalysis. *Trends in Biochemical Sciences*, 1996. 21(6): p. 220-224.

- (2) Line 105 and extended data fig. 5, the authors state that K-turns are a conserved feature of archaeal RNase Ps; several recent papers in the literature have made this assertion, and this is not acknowledged here, nor do authors provide adequate evidence to support the assertion on the basis of their findings.

We rewrote the K-turn section as “Consistent with previous predictions and structural data, phylogenetic and secondary structural analyses revealed that an extension with a K-turn beyond stem P12 is a conserved feature in most archaeal and eukaryotic RNase P RNAs” and cited the following papers [19-23].

19. Huang, L., S. Ashraf, and D.M.J. Lilley, The role of RNA structure in translational regulation by L7Ae protein in archaea. 2019. 25(1): p. 60-69.
20. Hall, T.A. and J.W. Brown, Archaeal RNase P has multiple protein subunits homologous to eukaryotic nuclear RNase P proteins. *RNA*, 2002. 8(3): p. 296-306.
21. Hartmann, E. and R.K. Hartmann, The enigma of ribonuclease P evolution. *Trends Genet*, 2003. 19(10): p. 561-9.
22. Andrews Andrew, J., A. Hall Thomas, and W. Brown James, Characterization of RNase P Holoenzymes from *Methanococcus jannaschii* and *Methanothermobacter thermoautotrophicus*, in *Biological Chemistry*. 2001. p. 1171.
23. Altman, S. and L.A. Kirsebom, *Ribonuclease P : The RNA world Second edition*. 1999, Cold Spring Harbor Laboratory Press, Cold Spring Harbor, NY. p. 351-378.

3. Lastly, the authors frequently attribute functional roles to structural features, without supporting functional evidence. For example, they often state that particular features “stabilize” some second feature of the structure; such an assertion would require evidence that without the first element, the second is not formed (e.g., line 136 – what is the evidence that the k-turn is not formed in the absence of Rpp38?; line 153; what is the evidence that the RPR needs to be stabilized by the proteins?). The authors should either provide evidence, or clarify that their functional attribution is speculation based on the model.

Thanks for this good point. Following this reviewer’s suggestion, we have revised the manuscript; we either deleted the statements that have no experimental evidence or we specifically stated that the functional attributions of structural features are speculation based on the model.

line 136 – what is the evidence that the K-turn is not formed in the absence of Rpp38?

We do not have experimental evidence whether K-turn is formed or not in the absence of L7Ae. We rephrased the manuscript as “L7Ae specifically recognizes the K-turn and helps stem P12.1 fold back onto P12.” (Page 12).

A recent study showed evidence that an RNA fragment with a K-turn sequence is unstructured in the absence of L7Ae binding, but folds on binding the protein (L7Ae is another name for archaeal Rpp38) [21].

21. Huang, L., S. Ashraf, and D.M.J. Lilley, The role of RNA structure in translational regulation by L7Ae protein in archaea. 2019. 25(1): p. 60-69.

line 153; what is the evidence that the RPR needs to be stabilized by the proteins?

When *M. jannaschii* RPR was *in vitro* transcribed and purified alone, RPR tended to form soluble aggregates as revealed by size exclusion chromatography analysis (Supplementary Fig. 1b). Notably, when purified L7Ae was added into the transcription reaction, RPR behaved properly as a monodispersed molecule, suggesting that L7Ae likely functions as a chaperon for the correct folding and/or stability of RPR (Supplementary Fig. 1b).

Additional comments:

4. There is generally insufficient discussion of how the structure and its components differ from the input models, including proteins, RPR (*Mja* v. *Pho*) and tRNA (*Tyr* v. *Phe*).

As shown in the response to Point #1, we revised manuscript and included detailed

description of model building and the differences between the final structures and their input model in the method section in revised manuscript.

5. Line82: The analogy to the flying pterosaur is cute, but given the C2 symmetry of the structure, misleading.

Following this reviewer's suggestion, we removed the analogy to flying pterosaur in the revised manuscript.

6. Page 5, lines 98 &103: there is no extended data fig. 4c

Thanks for pointing out this mistake. It should be extended data fig. 4b. The figures were reorganized in the revised manuscript. Extended Data Fig. 4b is Supplementary Fig. 6a in the revised manuscript.

7. Line 111: The descriptor of the proteins acting as a "extended hinge" is unclear and/or not supported by the data.

We rewrote the description of the proteins as "The protein components of the *Mja*RNase P complex are arranged into an elongated configuration, which can be divided into three submodules, one (Pop5-Rpp30)₂ heterotetramer at the centre and two Rpp21-Rpp29-L7Ae heterotrimers at the ends (Fig. 2b). These protein submodules are intimately stringed together and serve as a long, extended holder to accommodate two catalytic RPRs through a highly basic surface (~6,500 Å²) (Fig 2c). (Page7).

8. Page 6, line 118: It is unclear whether the data provides the resolution needed to see the mentioned base and side-chain interactions. The potential importance of the interaction is not supported by evidence.

Thanks for this point. We agree with the reviewer that the resolution of the EM density map cannot provides unambiguous details to support the side-chain interaction between G213 of RPR and Ile104 and Trp120 of Pop5. We removed this sentence in the revised manuscript.

9. Page 7, line 147, and figure 3b; it looks like the labels on the tRNA arms are swapped.

Thanks. We corrected this mistake in the revised manuscript (Fig. 6b).

10. line 186, figure 4, the authors don't justify inclusion of type A archaeal RNase P.

Thanks for this point. We removed this sentence in the revised manuscript. We rewrote this part as "The two RNA anchors respectively locate in the C and S domains of RPR, functioning as a 'measuring device' to recognize the coaxially stacked

acceptor and T ψ C arms of tRNA substrates, which measure a fixed distance of 12 base pairs in all tRNA molecules (Fig. 6a). This ‘double-RNA-anchor’ for tRNA recognition is conserved in most archaeal and bacterial RPRs. (Page 13).

11. line 204; functional equivalence of type M and type P RPR is unclear and/or not well justified.

Careful database search and analysis revealed some *P. aerophilum* tRNA genes (P type) do not have 3'-CCA sequence (Supplementary Fig. 15). Therefore, although *P. aerophilum* RNase P RPR contains a 3'-CCA recognition element in stem P15, it is not clear whether *P. aerophilum* RNase P employs the 3'-CCA base-pairing mechanism for substrate recognition. Given that type-P archaeal RPR lacks most of the S domain especially the T-loop anchor (CR-II/CR-III), it is an enigma how such a minimal version of RPR could process the tRNA in the absence of the protein components. To answer this question, structure of the type-P RNase P in complex with a tRNA substrate is required. In the revised manuscript, we removed Fig. 4 and the speculation about the tRNA recognition and processing mechanism by type-P archaeal RNase P.

12. line 211-213; the authors present no evidence supporting structural or evolutionary assertions about CCA recognition by eukaryal RNase P.

We have cited our recently papers on the cryo-EM structures of human and yeast RNase Ps in the revised manuscript [1] [2]. These studies support the notion that both the 3'-CCA recognition element and the A-anchor were lost in eukaryotic RNase Ps.

13. Figure 2 legend; the authors might consider “colored to match label”, instead of describing colors such as “slate”, “forest” and “deeptale”.

In the revised manuscript, we reorganized Fig. 2b as Fig. 3a, and we modified its figure legend as “Secondary structure of *Mja*RPR. Structural elements are colored to match their labels”.

14. Figure 2b, line 270; the dotted line shows the conserved, and well-known P4 pseudoknot, not tertiary contacts observed in the structure.

Thanks for pointing out this mistake. It is corrected in Fig. 3a legend in the revised manuscript.

15. Figure 3a; the star and triangle used to mark the recognition features are hard to see.

To better show the tRNA recognition features, we removed the star and triangle in the revised Fig. 6a. The details of tRNA recognition at the two anchor sites are shown in

the close-up views in Fig. 6b and 6c.

16. Figure 4 and extended figure 7 are not addressed in the body of the article.

In the revised manuscript, we removed Fig. 4 and the speculation about the tRNA recognition and processing mechanism by type-P archaeal RNase P. Please refer to our response to Point 11.

Original Extended Fig. 7 is reorganized in the revised manuscript.

Original Extended Fig. 7a becomes Fig. 5c, and is addressed in Page 12.

Original Extended Fig. 7b becomes Supplementary Fig. 12b, and is addressed in Page 10.

Original Extended Fig. 7c becomes Supplementary Fig. 12c, and is addressed in Page 12.

Original Extended Fig. 7d becomes Fig. 4d, and is addressed in Page 10.

17. “Tailor” when meaning “tail”?

Thanks for pointing out this typo. We have corrected this mistake in the revised manuscript.

18. The authors state that the active site geometry is “ideal” for binding two magnesium ions; it is not clear whether their data adequately constrain the model to make significant new conclusions.

We rewrote this section to explain why *Mja*RPR nucleotides at the active site likely play the same role in coordinating two Mg^{2+} ions essential for catalysis.

“The spatial arrangement of *Mja*RPR nucleotides around the 5’ end of tRNA, including G40, U41, A233 and A234 as well as the universally conserved uridine U42 in stem P4, highly resembles those observed in *T. maritima* and the newly reported yeast and human RNase P-tRNA complex structures (Fig. 7a). In the yeast RNase P structure in complex with a pre-tRNA substrate, equivalent nucleotides A91, U92, U93, G343 and A344 coordinates two catalytic Mg^{2+} ions (Fig. 7a). Consistent with this structural resemblance, single nucleotide deletion of U42 or replacement with an adenosine (U42A) greatly diminished the enzymatic activity of *Mja*RNase P, strongly supporting that *Mja*RPR nucleotides at the active site likely play the same role in coordinating two Mg^{2+} ions essential for catalysis (Fig. 7b). (Page 15).

Reviewer 2

1- Information about the precursor tRNA that was used, a figure showing its secondary structure would be informative. The best would be if the authors used a M.

jannaschii pre-tRNA.

We have performed tRNA processing activity assay using three different pre-tRNA substrates, human pre-tRNA^{Val} (without CCA), *Mja* pre-tRNA^{Arg} (without CCA) and *E. coli* pre-tRNA^{Tyr} (with CCA). Their secondary structures are shown in Supplementary Fig. 2a in the revised manuscript. Notably, *Mja*RNase P exhibited similar activity to all three substrates, suggesting that the absence of the 3' CCA sequence does not affect the cleavage activity of pre-tRNA by *Mja*RNase P (Supplementary Fig. 2b). We used *E. coli* pre-tRNA^{Tyr} in the *Mja*RNase P-tRNA complex structure study.

2- In Fig 1b the band representing the 5' matured tRNA is shown while the 5' leader is not seen. Please also show the mobility of the 5' leader. This is relevant in order to understand that the reconstituted RNase P cleave at the correct site. In this context, you need to include controls and one control is how the reconstituted RNase P perform in relation to native RNase P. Also, it is informative to follow the reaction over time both for the reconstituted RNase P and native RNase P.

Following this reviewer's suggestion, we include a tRNA 5' processing gel in Supplementary Fig. 2b, which includes bands of pre-tRNA, matured tRNA and 5' leader. Due to the special culturing conditions of *M. jannaschii* (2 atm of H₂/CO₂ (80 : 20 ratio); 85 °C), it is difficult to culture *M. jannaschii* cells in the laboratory environment. Therefore, we could not obtain native endogenous *Mja*RNase P holoenzyme. In the revised manuscript, we wrote "Whether the *Mja*RNase P holoenzyme adopts a dimeric conformation *in vivo* is still unclear and warrants further investigations." (Page 17).

3- What is the final Mg²⁺ concentration?

We reconstituted *Mja*RNase P in a buffer that contains 50 mM Hepes pH 7.5, 50 mM MgCl₂, 200 mM NaCl, 500 mM KCl, 50 mM potassium acetate. Therefore, the final Mg²⁺ concentration is 50 mM.

4- Do you have any information about the composition of the native RNase P? Does it appear as dimer in the cell?

Due to the special culturing conditions of *M. jannaschii* (2 atm of H₂/CO₂ (80 : 20 ratio); 85 °C), it is difficult to culture *M. jannaschii* cells in the laboratory environment. The composition of the *Mja*RNase P were identified based on their sequence similarities to the eukaryotic RNase P proteins [22-24].

Because of the same technical reason, we cannot obtain endogenous *Mja*RNase P holoenzyme, and therefore we do not know whether it appears as a dimer in *M. jannaschii* cells. In the revised manuscript, we wrote "Whether the *Mja*RNase P

holoenzyme adopts a dimeric conformation *in vivo* is still unclear and warrants further investigations.” (Page 17).

5- For the discussion about the function/mechanism of RNase P cleavage I recommend that you refer to previous reports. Much of the discussion is not new since it has been discussed/reported elsewhere. For example, the discussion about the interaction between the tRNA T-/D-loop region and RNase P, cleavage mechanism (involvement of two metal ion), interactions between the substrate and the RPR and the role of proteins, measuring mechanism etc. For references, see e.g. Liu F, Altman S. 2010. Ribonuclease P. Protein Reviews vol 10. Springer, New York Dordrecht Heidelberg, London, and chapters and references therein. After 2010 there has been reports relevant to the present study and I strongly recommend to go through these and take these publications into consideration.

Thanks for this good point. We cited the relevant papers in the revised manuscripts [4, 13, 25-31].

4. Reiter, N.J., et al., Structure of a bacterial ribonuclease P holoenzyme in complex with tRNA. *Nature*, 2010. 468(7325): p. 784-9.
13. Liu, X., Y. Chen, and C.A. Fierke, Inner-Sphere Coordination of Divalent Metal Ion with Nucleobase in Catalytic RNA. *J Am Chem Soc*, 2017. 139(48): p. 17457-17463.
25. Altman, S. and L.A. Kirsebom, Ribonuclease P : The RNA world Second edition. 1999, Cold Spring Harbor Laboratory Press, Cold Spring Harbor, NY. p. 351-378.
26. Yuan, Y. and S. Altman, Substrate recognition by human RNase P: identification of small, model substrates for the enzyme. *EMBO J*, 1995. 14(1): p. 159-68.
27. Liu, F. and S. Altman, Requirements for cleavage by a modified RNase P of a small model substrate. *Nucleic Acids Res*, 1996. 24(14): p. 2690-6.
28. Protein Reviews: Ribonuclease P. 1 ed. Ribonuclease P, ed. F. Liu and S. Altman. Vol. 10. 2010: Springer-Verlag New York. XVI, 283.
29. Sinapah, S., et al., Cleavage of model substrates by archaeal RNase P: role of protein cofactors in cleavage-site selection. *Nucleic Acids Res*, 2011. 39(3): p. 1105-16.
30. Chen, W.Y., et al., Fidelity of tRNA 5'-maturation: a possible basis for the functional dependence of archaeal and eukaryal RNase P on multiple protein cofactors. *Nucleic Acids Res*, 2012. 40(10): p. 4666-80.
31. Kimura, M., Structural basis for activation of an archaeal ribonuclease P RNA by protein cofactors. *Biosci Biotechnol Biochem*, 2017. 81(9): p. 1670-1680.

6- Since the 5' leader is not detected in the complex the structure represents the post-cleavage state and the data should be discussed in this perspective.

We rewrote a paragraph in the revised manuscript to specifically state that the tRNA substrate in the *Mja*RNase P-tRNA complex is the mature form without the 5' leader.

“To further understand how the tRNA substrate is recognized and processed by *Mja*RNase P, we mixed *Mja*RNase P with *E. coli* pre-tRNA^{Tyr} at a ratio of 1:10 and subjected the mixture to cryo-EM single particle analysis. Notably, the 5' leader of pre-tRNA^{Tyr} was cleaved during EM sample preparation, therefore obtained the three-dimensional reconstruction of *Mja*RNase P in complex with the mature form of the tRNA^{Tyr} substrate at a resolution of 4.3 Å (Fig. 2a, Supplementary Fig. 4 and Supplementary Table 2)”. (Page 7).

7- The Type M lacks residues in P15-loop that can pair with the pre-tRNA 3' end (3' NCC-motif) and the bound tRNA has a 3' CCA, but information whether the tRNA genes in *Methanocaldococcus jannaschii* encodes the 3' CCA needs to be included (see also my comment about the origin of the pre-tRNA above). If the tRNA genes do not encode CCA might provide a rationale to why residues that can pair with the pre-tRNA 3' end is absent.

Thanks for this good point. We performed database search and sequence analysis and found that 9 out of 35 *M. jannaschii* tRNA genes do not contain an 3'-RCCA sequence (Supplementary Fig. 15). This observation is in accordance with the fact that *M. jannaschii* RPR lacks a 3'-RCCA recognition element in its small terminal loop at stem P15 (Fig. 3a), suggesting that *Mja*RNase P might have lost this tRNA recognition element during evolution.

8- When you discuss the P type RPR I recommend that you also discuss the catalytic performance of the C-domain lacking the S-domain.

Careful database search and analysis revealed some *P. aerophilum* tRNA genes (P type) do not have 3'-CCA sequence (Supplementary Fig. 15). Therefore, although *P. aerophilum* RNase P RPR contains a 3'-CCA recognition element in stem P15, it is not clear whether *P. aerophilum* RNase P employs the 3'-CCA base-pairing mechanism for substrate recognition. Given that type-P archaeal RPR lacks most of the S domain especially the T-loop anchor (CR-II/CR-III), it is an enigma how such a minimal version of RPR could process the tRNA in the absence of the protein components. To answer this question, structure of the type-P RNase P in complex with a tRNA substrate is required. In the revised manuscript, we removed Fig. 4 and the speculation about the tRNA recognition and processing mechanism by type-P archaeal RNase P.

9- In the text please define “..the T-loops...” (e.g. line 135) so the reader clearly understand that you are not discussing the tRNA T-loop.

Thanks for this good point. In the revised manuscript, we defined the T-loop of RPR

as “The other two conserved regions CR-II and CR-III between stems P10 and P12 fold into two interleaved T-loops for tRNA substrate binding (Fig. 6).” (Page 8).

10- Line 29 “..one of only two ribozymes..”, what is the other ribozyme?

The other ribozyme is ribosome.

11- Lines 54-57 The sentence “Type A RNA that largely...” is misleading please revise.

We rewrote the sentence as: “A-type archaeal RPR largely resembles bacterial RPRs and displays trace amount of catalytic activity *in vitro*, whereas M-type RPR diverges more from bacterial RPRs with less complex structure and has not been shown the ability as an RNA-only ribozyme” in the revised manuscript (Page 4).

12- Line 189-192 Do you have any biochemical evidence to support this statement? Please clarify.

The ~ 4.0-Å resolution at the catalytic center in the *Mja*RNase P-tRNA complex structure suffices to show that the 5' leader of tRNA is absent in the catalytic center, indicating that the tRNA molecule in the complex is a mature tRNA product after cleavage (Fig. 6c). Recently, we determined the cryo-EM structure of yeast RNase P holoenzyme in complex with a pre-tRNA substrate [2]. In this structure, the 5' leader of pre-tRNA is clearly visible in the active site (Fig. 6d). Comparative analysis revealed that *Mja*Pop5 and its yeast homolog Pop5 occupy the same location on their respective RPRs and hold the zigzagged CR-IV of RPR in their deep basic clefts in the same manner (Fig. 6d). In the yeast RNase P-tRNA complex structure, Pop5 stabilizes CR-IV to make direct stacking interactions with nucleobases at the -1, -2 and -3 positions of the 5' leader of pre-tRNA (Fig. 6d). The close structural resemblance between *Mja*Pop5 and yeast Pop5 suggests that it is very likely *Mja*Pop5 employs the same mechanism to recognize the 5' leader of pre-tRNAs (Fig. 6c and 6d).

13- Line 199-200 There are exceptions to this in Bacteria, e.g., *Chlamydia* spp. and certain cyanobacteria.

Thanks. We rephrased the text as “...the 3'-CCA sequence of bacterial tRNAs is also recognized by a conserved RNA element, loop L15, **in most bacterial RPRs** through base-pairing interactions.” (Page 13).

14- Line 209-220 This is not the case since RNase P processing is affected when the interaction with the pre-tRNA 3' end and RNase P is disturbed. But this is dependent on the pre-tRNA structure.

As stated in response to Point 8, type-P archaeal RPR lacks most of the S domain especially the T-loop anchor (CR-II/CR-III). It is not clear how such a deletion version of RPR could process the tRNA in the absence of the protein components. To answer this question, structure of the type-P RNase P in complex with a tRNA substrate is required. In the revised manuscript, we removed Fig.4 and the speculation about the tRNA recognition and processing mechanism by type-P archaeal RNase P. We removed lines 209-213 in revised manuscript.

15- Line 312-313 How did you removed the protein contaminations after heat treatment?

Because most protein contaminations precipitated under heat treatment, we removed those contaminants by centrifugation. The proteins were further purified by ion exchange and size exclusion chromatography steps.

16- Line 320 What was the rational to add Rpp38? Please clarify.

When *M. jannaschii* RPR was *in vitro* transcribed and purified alone, RPR tended to form soluble aggregates as revealed by size exclusion chromatography analysis (Supplementary Fig. 1b). Notably, when purified L7Ae (another name of Rpp38) was added into the transcription reaction, RPR behaved properly as a monodispersed molecule, suggesting that L7Ae likely functions as a chaperon for the correct folding and/or stability of RPR (Supplementary Fig. 1b).

17- Line 326 Please define KoAc?

We have replaced KoAc with potassium acetate in the revised manuscript.

18- Extended Data Figure 3 legend. Please included references for the crystal structures of *P. horikoshii* proteins.

We have included the corresponding references in the revised manuscript.

19- Extended Data Figure 6 legend. Please expand the text after (d), e.g., how do you define ideal?

Thanks for this point. In the revised manuscript we include a separate section “Dimeric organization of *Mja*RNase P” and a figure (Fig. 8) with new experimental data to address why the dimerization of *Mja*RNase P is important for the structural stability and the enzymatic activity of *Mja*RNase P. (Pages 15-17).

20- Extended Data Figure 8 legend. *T. maritime* should be *T. maritima* (also in Extended Data Figure 9 legend). Also, give reference Reiter et al. 2010.

We have corrected the typo and included the corresponding references in the revised manuscript.

21- Extended Data Figure 10 legend. Include reference 16 in the figure legend text. Change *Pyrococcus horikoshii* to *P. horikoshii*.

Corrected.

Reviewer 3

Major comments

1- The authors suggest that the tRNA 3' CCA is not recognized with sequence-specificity. This would be an important conclusion of the study and should thus be supported by the structural data. The authors should show the density for the 3'CCA and its RPR interaction. For example, if the path of the RNA backbone were ambiguous, the CCA bases might instead interact with the RPR through triplet-helix interactions. At 4.3Å such details may be difficult to observe for nucleic acid in particular, depending on the local resolution.

Thanks for this good point. We carefully reexamined the EM density of the 3' terminus of tRNA. After the G1-C81 base pair, only two nucleotides A82 and C83 can be modeled into the density (Supplementary Fig. 16). Therefore, two nucleotides C84 and A85 in the 3'-CCA motif are not visible in the EM density, presumably disordered in the *Mja*RNase P-tRNA complex structure. In addition, the structure of stem P15 shows that the local space is too small to accommodate a triplet-helix interaction. Together, we conclude that it is unlikely the 3'-CCA is recognized by *Mja*RPR through base-pair interactions as in bacterial RNase Ps.

2- Related to point 1, while sequence-specific recognition of the CCA element may not be required, its presence may be important for cooperative binding of the tRNA in addition to the T and A acceptor loop interaction. Do the authors have data on how sequence changes or absence of the CCA element affect the in vitro binding affinity/cleavage activity of pre-tRNA to RNase P? If there is no clear data on this, the authors should reflect this in their model (text and Fig 4) that the presence of the CCA may still be needed for pre-tRNA specificity, even if the sequence is not read out directly.

We have performed tRNA processing activity assay using three different tRNA substrates, human tRNA^{Val} (without CCA), *Mjat*RNA^{Arg} (without CCA) and bacterial tRNA^{Tyr} (with CCA). As shown in Supplementary Fig. 2b in the revised manuscript, *Mja*RNase P exhibited similar activity to all three substrates, suggesting that the absence of the 3' CCA does not affect the cleavage activity of pre-tRNA by *Mja*RNase P.

Database search and sequence analysis revealed that some *M. jannaschii* pre-tRNAs do not contain an CCA sequence at their 3' termini (Supplementary Fig. 15). This observation is in accordance with the fact that *Mja*RPR lacks a CCA recognition element in its small terminal loop at stem P15 (Fig. 3a), suggesting that *Mja*RNase P might have lost this tRNA recognition element during evolution.

This should also be reflected in the Abstract.

Thanks for this point. The Abstract was modified accordingly.

3- Can the authors speculate how RNase P dimerization might influence cleavage activity, as this seems to be a conserved feature?

The (Pop5-Rpp30)₂ heterotetramer binds to the C domain of both RPR molecules symmetrically in the *Mja*RNase P dimeric complex, so that Rpp30 is involved in tRNA binding in one monomeric complex while sitting on the short P3 stem of RPR from the other complex (Fig. 8b). Markedly, the equivalent Rpp30 that contact the tRNA substrate in both human and yeast monomeric RNase P complexes is also buttressed by additional protein subunit and/or RPR element (Fig. 8c). Structurally, these interactions appear to stabilize Rpp30 for complex assembly and tRNA substrate binding (Fig. 8b and 8c). Consistent with this idea, alanine substitution of *P. horikoshii* Rpp30 Lys196 (equivalent to *Mja*Rpp30 Lys198) at the interface between Rpp30 and the second RPR in the dimer greatly reduced the pre-tRNA cleavage activity, suggesting that the dimeric conformation of archaeal RNase P is crucial for its enzymatic activity (Fig. 8b) [8]. To further examine the function of dimerization, we designed a monomeric mutant *Mja*RPR with an artificial P19 stem inserted between nucleotides C222 and G223, which should preclude the dimer formation (Supplementary Fig. 18). Indeed, *in vitro* reconstitution with this mutant RPR resulted in a monomeric *Mja*RNase P complex as revealed by both gel filtration and negative staining EM analyses (Fig. 8d and 8e). *In vitro* activity assay showed that this monomeric mutant *Mja*RNase P exhibited substantially reduced pre-tRNA processing activity (Fig. 8f), underscoring the importance of dimerization in the *in vitro* activity of *Mja*RNase P.

4- The Abstract is misleading to the nature of the 3' CCA recognition (Line 44) in this structure. The authors should rephrase this to clarify that the 3'CCA element recognition may not be important for M-type RNase P enzymes.

We have rephrased the Abstract following this reviewer's suggestion.

Minor comments:

1. It is difficult to follow the RNA elements between figures 2a/b and 3 due to the

180 degree rotation of the structure. If the authors wish to keep this arrangement, please improve the labeling of Figure 3a to include: tRNA acceptor and TyC arms, modules A/B, tRNA 5' and 3' ends, and the RPR elements.

Thanks for this good point. In revised manuscript, we redrew Figs. 6a, 6b and 6c so that they have the same orientation as Fig. 3c, and that the structural elements of RPR are at the same locations in all figures.

2. The authors should correct the text for typos and use of grammar, as I spotted a few errors throughout.

We have revised the manuscript and fixed the typo and grammar problems.

3. Please mention in the main text that C2 symmetry was used to improve the resolution, even though it is already stated in the methods.

We mentioned the use of the C2 symmetry to improve the resolution in the revised main text.

“The resolutions of the cryo-EM reconstruction were substantially improved by applying the two-fold symmetry, suggestive of a very rigid dimeric interface in the *MjaRNase P* complex” (Page 7).

4. Line 30: Please correct to "only one of two ribozymes present"

Corrected.

5. Line 72: It would be useful to mention here that the pre-tRNA was cleaved during sample preparation, since this otherwise might lead to confusion with the line above (Line 70: "[...] *MjRNase P* and a pre-tRNA^{Tyr} substrate at a ratio of 1:10 [...]").

Thanks for this point. We rewrote this section as “To further understand how the tRNA substrate is recognized and processed by *MjaRNase P*, we mixed *MjaRNase P* with *E. coli* pre-tRNA^{Tyr} at a ratio of 1:10 and subjected the mixture to cryo-EM single particle analysis. Notably, the 5' leader of pre-tRNA^{Tyr} was cleaved during EM sample preparation. We obtained the three-dimensional reconstruction of *MjaRNase P* in complex with the mature form of the tRNA^{Tyr} substrate at a resolution of 4.3 Å (Fig. 2a, Supplementary Fig. 4 and Supplementary Table 2)”. (Page 7).

6. Line 145: Please repeat here that pre-tRNA^{Tyr} is being used in the co-complex. Related to this, is the distance between the T and Acceptor stems invariant in all tRNAs, and is this worth noting in the text?

Following this reviewer's suggestion, we rewrote this section as “Although

pre-tRNA^{Tyr} was used in the cryo-EM analysis of the *Mja*RNaseP-tRNA^{Tyr} complex, the ~ 4.0-Å resolution at the catalytic center suffices to show that the 5' leader of tRNA is absent in the structure, indicating that the tRNA molecule in the complex is the mature tRNA product after cleavage (Supplementary Fig. 13a)". (Page 12).

The distance between the T ψ C and acceptor stems in all tRNAs is invariant – 12 base-pairs or ~ 45 Å. We specifically mentioned this point in the revised manuscript as “The two RNA anchors respectively locate in the C and S domains of RPR, functioning as a ‘measuring device’ to recognize the coaxially stacked acceptor and T ψ C arms of tRNA substrates, which measure a fixed distance of 12 base pairs (~ 45 Å) in all tRNA molecules (Fig. 6a)". (Page 13).

7. For Figures 3a, c, d, please indicate the site of cleavage the 5' end of the tRNA or indicate this in the Figure legend.

We labeled the cleavage site of the 5' end of tRNA in Fig. 6a, 6c and 6d in the revised manuscript.

8. In Figure 4, please shorten the RPR length in the P-type RNase P model by 2/3 to more accurately represent its unique architecture.

Careful database search and analysis revealed some *P. aerophilum* tRNA genes (P type) do not have 3'-CCA sequence (Supplementary Fig. 15). Therefore, although *P. aerophilum* RNase P RPR contains a 3'-CCA recognition element in stem P15, it is not clear whether *P. aerophilum* RNase P employs the 3'-CCA base-pairing mechanism for substrate recognition. Given that type-P archaeal RPR lacks most of the S domain especially the T-loop anchor (CR-II/CR-III), it is an enigma how such a minimal version of RPR could process the tRNA in the absence of the protein components. To answer this question, structure of the type-P RNase P in complex with a tRNA substrate is required. In the revised manuscript, we removed Fig. 4 and the speculation about the tRNA recognition and processing mechanism by type-P archaeal RNase P.

9. Do the authors find a tRNA-free population of RNase P in their 3D classification, and if so, does this yield any insights into the tRNA binding mechanism?

We did not find a tRNA-free population of RNase P in the 3D classification. We mixed *Mja*RNase P with the tRNA substrate at a ratio of 1:10, so that *Mja*RNase P was saturated by tRNA. To address reviewer's question, we determined the apo structure of *Mja*RNase P at 4.6 Å resolution, which is almost identical to the tRNA-bound *Mja*RNase P structure (Fig. 2a, 2b and 2d), suggesting that the *Mja*RNase P holoenzyme adopts a stable conformation that is optimal for tRNA binding.

10. To better judge the density quality, it would be helpful to include fits of the density with the proteins and RNAs in the supplement. Zoom-ins of important regions would be helpful too.

We have included fits of the EM density with proteins and RNAs in Supplementary Fig. 5 in the revised manuscript. Zoom-ins of important regions are shown in Fig. 8a and Supplementary Figs. 10a, 10b, 12a, 13a, 13b, 13c and 16.

1. Wu, J., et al., *Cryo-EM Structure of the Human Ribonuclease P Holoenzyme*. Cell, 2018. **175**(5): p. 1393-1404.e11.
2. Lan, P. and M. Tan, *Structural insight into precursor tRNA processing by yeast ribonuclease P*. 2018. **362**(6415).
3. Torres-Larios, A., et al., *Crystal structure of the RNA component of bacterial ribonuclease P*. Nature, 2005. **437**(7058): p. 584-7.
4. Reiter, N.J., et al., *Structure of a bacterial ribonuclease P holoenzyme in complex with tRNA*. Nature, 2010. **468**(7325): p. 784-9.
5. Oshima, K., et al., *Structural basis for recognition of a kink-turn motif by an archaeal homologue of human RNase P protein Rpp38*. Biochem Biophys Res Commun, 2016. **474**(3): p. 541-546.
6. Oshima, K., et al., *Crystal structures of the archaeal RNase P protein Rpp38 in complex with RNA fragments containing a K-turn motif*. Acta Crystallogr F Struct Biol Commun, 2018. **74**(Pt 1): p. 57-64.
7. Hamma, T. and A.R. Ferré-D'Amaré, *Structure of Protein L7Ae Bound to a K-Turn Derived from an Archaeal Box H/ACA sRNA at 1.8 Å Resolution*. Structure, 2004. **12**(5): p. 893-903.
8. Kawano, S., et al., *Crystal structure of protein Ph1481p in complex with protein Ph1877p of archaeal RNase P from Pyrococcus horikoshii OT3: implication of dimer formation of the holoenzyme*. J Mol Biol, 2006. **357**(2): p. 583-91.
9. Honda, T., et al., *Structure of an archaeal homolog of the human protein complex Rpp21-Rpp29 that is a key core component for the assembly of active ribonuclease P*. J Mol Biol, 2008. **384**(3): p. 652-62.
10. Demeshkina, N., et al., *A new understanding of the decoding principle on the ribosome*. Nature, 2012. **484**(7393): p. 256-9.
11. Kazantsev, A.V. and N.R. Pace, *Bacterial RNase P: a new view of an ancient enzyme*. Nat Rev Microbiol, 2006. **4**(10): p. 729-40.
12. Steitz, T.A. and J.A. Steitz, *A general two-metal-ion mechanism for catalytic RNA*. Proc Natl Acad Sci U S A, 1993. **90**(14): p. 6498-502.
13. Liu, X., Y. Chen, and C.A. Fierke, *Inner-Sphere Coordination of Divalent Metal Ion with Nucleobase in Catalytic RNA*. J Am Chem Soc, 2017. **139**(48): p. 17457-17463.
14. Kirsebom, L.A. and S. Trobro, *RNase P RNA-mediated cleavage*. IUBMB Life, 2009. **61**(3): p. 189-200.
15. Perreault, J.P. and S. Altman, *Important 2'-hydroxyl groups in model substrates for M1 RNA, the catalytic RNA subunit of RNase P from Escherichia coli*. J Mol Biol, 1992. **226**(2): p. 399-409.

16. Fedor, M.J., *The role of metal ions in RNA catalysis*. *Curr Opin Struct Biol*, 2002. **12**(3): p. 289-95.
17. Hsieh, J., et al., *A divalent cation stabilizes the active conformation of the B. subtilis RNase P x pre-tRNA complex: a role for an inner-sphere metal ion in RNase P*. *J Mol Biol*, 2010. **400**(1): p. 38-51.
18. Christian, E.L., et al., *The P4 metal binding site in RNase P RNA affects active site metal affinity through substrate positioning*. *RNA*, 2006. **12**(8): p. 1463-7.
19. Beebe, J.A., J.C. Kurz, and C.A. Fierke, *Magnesium Ions Are Required by Bacillus subtilis Ribonuclease P RNA for both Binding and Cleaving Precursor tRNAAsp*. *Biochemistry*, 1996. **35**(32): p. 10493-10505.
20. Scott, W.G. and A. Klug, *Ribozymes: structure and mechanism in RNA catalysis*. *Trends in Biochemical Sciences*, 1996. **21**(6): p. 220-224.
21. Huang, L., S. Ashraf, and D.M.J. Lilley, *The role of RNA structure in translational regulation by L7Ae protein in archaea*. 2019. **25**(1): p. 60-69.
22. Hall, T.A. and J.W. Brown, *Archaeal RNase P has multiple protein subunits homologous to eukaryotic nuclear RNase P proteins*. *RNA*, 2002. **8**(3): p. 296-306.
23. Hartmann, E. and R.K. Hartmann, *The enigma of ribonuclease P evolution*. *Trends Genet*, 2003. **19**(10): p. 561-9.
24. Andrews Andrew, J., A. Hall Thomas, and W. Brown James, *Characterization of RNase P Holoenzymes from Methanococcus jannaschii and Methanothermobacter thermoautotrophicus*, in *Biological Chemistry*. 2001. p. 1171.
25. Altman, S. and L.A. Kirsebom, *Ribonuclease P : The RNA world Second edition*. 1999, Cold Spring Harbor Laboratory Press, Cold Spring Harbor, NY. p. 351-378.
26. Yuan, Y. and S. Altman, *Substrate recognition by human RNase P: identification of small, model substrates for the enzyme*. *EMBO J*, 1995. **14**(1): p. 159-68.
27. Liu, F. and S. Altman, *Requirements for cleavage by a modified RNase P of a small model substrate*. *Nucleic Acids Res*, 1996. **24**(14): p. 2690-6.
28. *Protein Reviews: Ribonuclease P*. 1 ed. Ribonuclease P, ed. F. Liu and S. Altman. Vol. 10. 2010: Springer-Verlag New York. XVI, 283.
29. Sinapah, S., et al., *Cleavage of model substrates by archaeal RNase P: role of protein cofactors in cleavage-site selection*. *Nucleic Acids Res*, 2011. **39**(3): p. 1105-16.
30. Chen, W.Y., et al., *Fidelity of tRNA 5'-maturation: a possible basis for the functional dependence of archaeal and eukaryal RNase P on multiple protein cofactors*. *Nucleic Acids Res*, 2012. **40**(10): p. 4666-80.
31. Kimura, M., *Structural basis for activation of an archaeal ribonuclease P RNA by protein cofactors*. *Biosci Biotechnol Biochem*, 2017. **81**(9): p. 1670-1680.

REVIEWERS' COMMENTS:

Reviewer #2 (Remarks to the Author):

The revised manuscript by Wan et al. has been improved and as stated in my original revision this contribution is interesting and provides insight into the structure of an archaeal (type M) RNase P. My comments have been addressed however some more comments that needs attention.

Line 55 insert are after that, should read "...are present..."

Line 67 include the following references Green et al. Nucl Acids Res 24, 1497-1503 (1996) and Loria and Pan Biochemistry 38, 8612-20 (1999). Note also that E. coli RNase P RNA is catalytically active at low concentrations of Mg in the presence of spermidine (Guerrier-Takada et al. Biochemistry 25 1509-15 (1986). Therefore the "...high ionic strength conditions..." should be carefully used.

Line 127-135 In Fig S2b there is an extra band in cleavage of pre-tRNA^{Arg}, which is not present without RNase P or with EDTA. Please comment on this.

Line 145 the sentence beginning with "Notablepreparation, we obtained...". Please clarify.

Line 181-182 insert P4 after "...motif (P4), occupying..." for clarity.

Line 369 Fig 8e. The positioning of the 5' leader should be shown to inform the reader that the dimeric and monomeric RNase P cleave at the same site.

Line 384 insert "the" between in and majority so it reads "...in the majority..."

Line 435 and elsewhere in the Methods section change 65 oC to 65oC.

Line 458 The sentence "Ten-fold..." it is not clear what you mean, ten-fold in relation to what?. Please revise.

Lines 481-482 "...200 mM salt..." Salt, which salt? Please clarify.

Line 547 Second Mja should also be in italic.

Line 658 Which pre-tRNA? Please specify.

Reference list

Check the reference list in particular with respect "journal abbreviations", e.g., Ref 4, 18, 23, 24, 30, 36, 42, 53, 54, 66 and 69.

Fig 7a Check whether one of the oxygens on U42 is marked as an amine, it should be red marked and on the figure I have it looks as if it is marked in blue.

Supplementary Fig 9 P. horikoshill should be P. horikoshii.

Supplementary Fig 15 Please state the full name of P. aerophilum.

Supplementary Fig 20 Insert space between Fig. and 3a should read "...Fig. 3a."

Reviewer #3 (Remarks to the Author):

I would like to commend the authors for their thorough revision, implementing changes in text, figures and providing additional data. The comparisons to yeast and human RNase P enzymes also add important information. I am happy to support publication of the manuscript in its current form.

Point-to-point responses to reviewers' comments

Cryo-EM structure of an archaeal ribonuclease P holoenzyme in complex with tRNA

Reviewer 2

The revised manuscript by Wan et al. has been improved and as stated in my original revision this contribution is interesting and provides insight into to the structure of an archaeal (type M) RNase P.

Thanks!

1. Line 55 insert are after that, should read "...are present..."

Corrected.

2. Line 67 include the following references Green et al. Nucl Acids Res 24, 1497-1503 (1996) and Loria and Pan Biochemistry 38, 8612-20 (1999). Note also that E. coli RNase P RNA is catalytically active at low concentrations of Mg in the presence of spermidine (Guerrier-Takada et al. Biochemistry 25 1509-15 (1986). Therefore the "...high ionic strength conditions..." should be carefully used.

Thanks for this good point. We rewrote this part as "It has been found that bacterial RPR even the C domain alone possesses catalytic activity under high ionic strength condition or in the presence of spermidine *in vitro*^{12, 13}. But the sole protein subunit RPP is essential for enhancing the efficiency and fidelity of substrate recognition and cleavage under physiological conditions^{14, 15, 16, 17, 18, 19}." We have included the references recommended by the reviewer in the revised manuscript.

3. Line 127-135 In Fig S2b there is an extra band in cleavage of pre-tRNA^{Arg}, which is not present without RNase P or with EDTA. Please comment on this.

It is likely that during the preparation of pre-tRNA^{Arg}, a fragment of pre-tRNA^{Arg} that contains the 5'-leader was generated and after cleavage by RNase P, this fragment gave rise to an extra band that is not present without RNase P or with EDTA. We labeled these extra bands in the revised Fig. S2b.

b

5. Line 145 the sentence beginning with "Notablepreparation, we obtained...". Please clarify.

We rewrote the sentence as “Notably, we obtained the three-dimensional reconstruction of *Mja* RNase P in complex with the mature form of the tRNA^{Tyr} substrate at a resolution of 4.3 Å. It is likely that the 5'-leader of the pre-tRNA^{Tyr} was cleaved during EM sample preparation.”

6. Line 181-182 insert P4 after "...motif (P4), occupying...." for clarity.

Corrected.

7. Line 369 Fig 8e. The positioning of the 5' leader should be shown to inform the reader that the dimeric and monomeric RNase P cleave at the same site.

We modified Fig 8e in the revised manuscript as suggested to show the 5'-leader.

8. Line 384 insert "the" between in and majority so it reads "...in the majority..."

Corrected.

9. Line 384 insert "the" between in and majority so it reads "...in the majority..."

Corrected.

10. Line 435 and elsewhere in the Methods section change 65 oC to 65°C.

Corrected.

11. Line 458 The sentence "Ten-fold...." it is not clear what you mean, ten-fold in relation to what?. Please revise.

We have rewrote the sentence as "The *Mja*RNase P was mixed with *E.coli* pre-tRNA^{Tyr} in a molar ratio of 1:10 to generate the *Mja*RNase P-pre-tRNA complex samples for EM experiments."

12. Lines 481-482 "...200 mM salt..." Salt, which salt? Please clarify.

We rewrote the sentence as "Since the sample was in a high-salt buffer, which hinders the cryo-EM data acquisition, the high-salt buffer was exchanged to a buffer that contains 200 mM salt (40 mM Hepes pH 8.0, 200 mM NaCl) on the grids quickly before plunge."

13. Line 547 Second *Mja* should also be in italic.

Corrected.

14. Line 658 Which pre-tRNA? Please specify.

We specified the pre-tRNA as pre-tRNA^{Phe} in the revised manuscript.

15. Check the reference list in particular with respect "journal abbreviations", e.g., Ref 4, 18, 23, 24, 30, 36, 42, 53, 54, 66 and 69.

Corrected.

16. Fig 7a Check whether one of the oxygens on U42 is marked as an amine, it should be red marked and on the figure I have it looks as if it is marked in blue.

The oxygen atom is in red. However, due to the shading effect in Fig. 7a, it looks as if it is marked in blue. We modified the color so that it appears as a red oxygen atom.

17. Supplementary Fig 9 P. horikoshill should be P. horikoshii.

Corrected.

18. Supplementary Fig 15 Please state the full name of P. aerophilum.

Corrected.

19. Supplementary Fig 20 Insert space between Fig. and 3a should read "..Fig. 3a."

Corrected.

Reviewer 3

I would like to commend the authors for their thorough revision, implementing changes in text, figures and providing additional data. The comparisons to yeast and human RNase P enzymes also add important information. I am happy to support publication of the manuscript in its current form.

Thanks!